# The regulatory impact of RNA-binding proteins on microRNA targeting

Sukjun Kim [1,11], Soyoung Kim[2,11], Hee Ryung Chang[1,11], Doyeon Kim[1,11], Junehee Park[1], Narae Son[1], Joori Park [3,4], Minhyuk Yoon[2], Gwangung Chae[2], Young-Kook Kim[5], V. Narry Kim [1,6], Yoon Ki Kim [3,4], Jin-Wu Nam [7], Chanseok Shin[2,8,9 ✉] & Daehyun Baek[1,10 ✉]

Argonaute is the primary mediator of metazoan miRNA targeting (MT). Among the currently identified >1,500 human RNA-binding proteins (RBPs), there are only a handful of RBPs known to enhance MT and several others reported to suppress MT, leaving the global impact of RBPs on MT elusive. In this study, we have systematically analyzed transcriptome-wide binding sites for 150 human RBPs and evaluated the quantitative effect of individual RBPs on MT efficacy. In contrast to previous studies, we show that most RBPs significantly affect MT and that all of those MT-regulating RBPs function as MT enhancers rather than suppressors, by making the local secondary structure of the target site accessible to Argonaute. Our findings illuminate the unappreciated regulatory impact of human RBPs on MT, and as these RBPs may play key roles in the gene regulatory network governed by metazoan miRNAs, MT should be understood in the context of co-regulating RBPs.

[1] School of Biological Sciences, Seoul National University, Seoul, Republic of Korea. [2] Department of Agricultural Biotechnology, Seoul National University, Seoul, Republic of Korea. [3] Creative Research Initiatives Center for Molecular Biology of Translation, Korea University, Seoul, Republic of Korea. [4] Division of Life Sciences, Korea University, Seoul, Republic of Korea. [5] Department of Biochemistry, Chonnam National University Medical School, Hwasun, Jeollanam-do, Republic of Korea. [6] Center for RNA Research, Institute for Basic Science, Seoul, Republic of Korea. [7] Department of Life Science, College of Natural Sciences, Hanyang University, Seoul, Republic of Korea. [8] Research Institute of Agriculture and Life Sciences, and Plant Genomics and Breeding Institute, Seoul National University, Seoul, Republic of Korea. [9] Research Center for Plant Plasticity, Seoul National University, Seoul, Republic of Korea. [10] Bioinformatics Institute, Seoul National University, Seoul, Republic of Korea. [11]These authors contributed equally: Sukjun Kim, Soyoung Kim, Hee Ryung Chang, Doyeon Kim. ✉email: cshin@snu.ac.kr; baek@snu.ac.kr

t is broadly accepted that metazoan miRNA targeting (MT) is governed by a ternary interaction of Argonaute (AGO), miRNA, and its mRNA target[1–4]. While AGO is the main mediator of MT, there is an increasing amount of evidence that other RNA-binding proteins (RBPs) also play important regulatory roles in MT[5]. For instance, phosphorylated Pumilio protein opens up the local hairpin structure of the *CDKN1B* 3′ UTR, where the miR-221/222 target site is otherwise inaccessible, and lead to productive MT by miR-221/222[6]. There are a few other examples of such an MT enhancer as PCBP2 and FUS[7,8]. In contrast, Dnd1, RBM38, and IGF2BP1 have been reported to function as MT suppressors where these RBPs suppress MT mostly by making the miRNA target site more inaccessible to AGO[9–11]. HuR and PTBP have been reported to be enhancers of MT by either recruiting AGO or opening the secondary structure to increase accessibility to AGO as well as suppressors of MT by competing against AGO[12–14].

On the other hand, >1,500 human RBPs have been identified[15] and each RBP is estimated to have on average 22,000 3′UTR-binding sites[16], leading to >33 million interactions that can occur between human RBPs and 3′UTRs (see below). Despite the enormous number of possible interactions, only a handful of aforementioned interactions have been examined so far, leaving almost all other interactions unexamined. Accordingly, our current understanding towards the global regulatory impact of RBPs on MT remains severely limited.

In this study, by analyzing the transcriptome-wide binding sites for 150 human RBPs and large-scale datasets that monitored the whole-transcriptome response to ectopically introduced or deleted miRNAs, we attempted to systematically evaluate the quantitative effect of RBPs on MT and thus to help gain a comprehensive insight into the gene regulatory network of metazoan miRNAs and their co-regulating RBPs.

## Results

**RBPs have a large number of 3′UTR-binding sites**. To accurately detect binding sites of RBPs, CLIP-seq (crosslinking and immunoprecipitation followed by sequencing) has been developed[17–22]. Taking advantage of this powerful technology, the ENCODE consortium has published a massive-scale dataset of an enhanced version, termed eCLIP-seq[16,23]. We obtained and analyzed the ENCODE eCLIP-seq dataset to identify the transcriptome-wide binding sites for 150 RBPs profiled in HepG2 and K562 cell lines.

Our analysis indicated that human RBPs not only bind to the 5′UTRs and coding regions, but also to the 3′UTRs substantially: the evaluated 150 RBPs have on average 22,000 3′UTR-binding sites (Supplementary Fig. 1a) ranging from 1,000 to 73,000. Extrapolating the average number of 3′UTR-binding sites to 1,500 human RBPs, we estimate that >33 million interactions, several orders of magnitude larger than the previously evaluated interactions, can occur between human RBPs and 3′UTRs.

Because metazoan MT occurs primarily in the cytoplasm[24,25], we quantitatively assessed the subcellular localization of RBPs by analyzing immuno-fluorescence images[26]. When measuring the cytoplasmic fraction compared to the nucleus fraction for each RBP, 91% of the evaluated RBPs exhibited >5% of cytoplasmic fraction (Supplementary Fig. 1b), suggesting that almost all RBPs are localized in the cytoplasm to a detectable degree. These findings demonstrate that human RBPs have a large number of 3′ UTR-binding sites and also localized in the cytoplasm despite the remarkably varying fraction, justifying our hypothesis to evaluate whether RBPs may globally influence MT.

**Strong association between RBP binding and enhanced miRNA targeting**. To examine the association between RBP binding and MT, we have generated a large dataset of mRNA-seq that measured the whole-transcriptome response to over-expressed miRNAs in HepG2 cell line. By combining this dataset with the ENCODE RBP-binding site (RBS) dataset, we collected target mRNAs with a single 7, 8mer miRNA target site (MTS) and tested whether the distance between an MTS and the nearest RBS on the 3′UTR, denoted as $d_{MTS-RBS}$, is correlated with MT efficacy (Fig. 1a, Supplementary Discussion), hypothesizing that the RBS located close to an MTS might influence MT. Although we examined whether $d_{MTS-RBS}$ is associated with MT efficacy either positively or negatively to potentially discover both MT enhancers and suppressors, the shorter $d_{MTS-RBS}$ was significantly correlated with the improved MT efficacy only (Fig. 1b, HepG2 panel). This association was significant even after correcting for known confounding factors (local AU content, 3′UTR size, target-site abundance, and seed-pairing stability)[27–34] by the multiple linear regression (Fig. 1c, HepG2 panel). To rule out these confounding effects more definitively, we selected a group of 3′UTRs that have different $d_{MTS-RBS}$ but have statistically indistinguishable confounding factors. Even after such a rigorous correction, our observation was still consistent and we were able to confirm the independent correlation between $d_{MTS-RBS}$ and MT efficacy on a global scale (Fig. 1d, HepG2 panel). When compared with known determinants of MT such as target-site abundance[29], the overall impact of $d_{MTS-RBS}$ on MT was significantly stronger than those of previously reported determinants, emphasizing its potential role as a key determinant of MT (Supplementary Fig. 1c).

To confirm our observation is not limited to HepG2 cell line, we analyzed a large dataset that monitored the whole-transcriptome response to overexpressed miRNAs in various other cell lines[29] (Fig. 1b–d). For the RBS information of these cell lines, we used the ENCODE RBS information obtained from both HepG2 and K562 cell lines, since we confirmed that RBSs are robust enough to be preserved between these two cell lines and thus these RBSs can also be applied to other cell lines (Supplementary Fig. 1d, e). Accordingly, we again observed a significant correlation between $d_{MTS-RBS}$ and MT efficacy with other independent datasets (Fig. 1b–d and Supplementary Fig. 1f) and with the preserved binding sites (Supplementary Fig. 1g), supporting that the observed association is general enough to be extended in various biological contexts.

However, given that these results are based on the transcriptome response to ectopically introduced miRNAs, it was crucial to confirm the results in an endogenous condition as well. To do so, we used a dataset of *DROSHA* and *DICER* knockout cell lines where endogenous miRNAs are depleted[35]. In accord with our previous results, we observed a significant stronger derepression of mRNAs with shorter $d_{MTS-RBS}$ (Fig. 1e and Supplementary Fig. 1h), demonstrating that the observed effect of RBPs on MT is a phenomenon occurring in an endogenous environment.

**Most RBPs are associated with enhanced miRNA targeting**. To examine if this correlation between $d_{MTS-RBS}$ and MT efficacy is observed with individual RBPs, we performed a correlation analysis for each individual RBP after correcting for confounding factors in various cell lines (Fig. 2a–c). First, 86%, 93%, and 94% of the RBPs evaluated in HepG2, HeLa, and other human cancer cell lines, respectively, exhibited significant correlations between $d_{MTS-RBS}$ and MT efficacy. These results illustrate that the regulatory impact of RBPs on MT may be broad in contrast to the previously reported examples[6–8,10–12,36]. Second, similar to

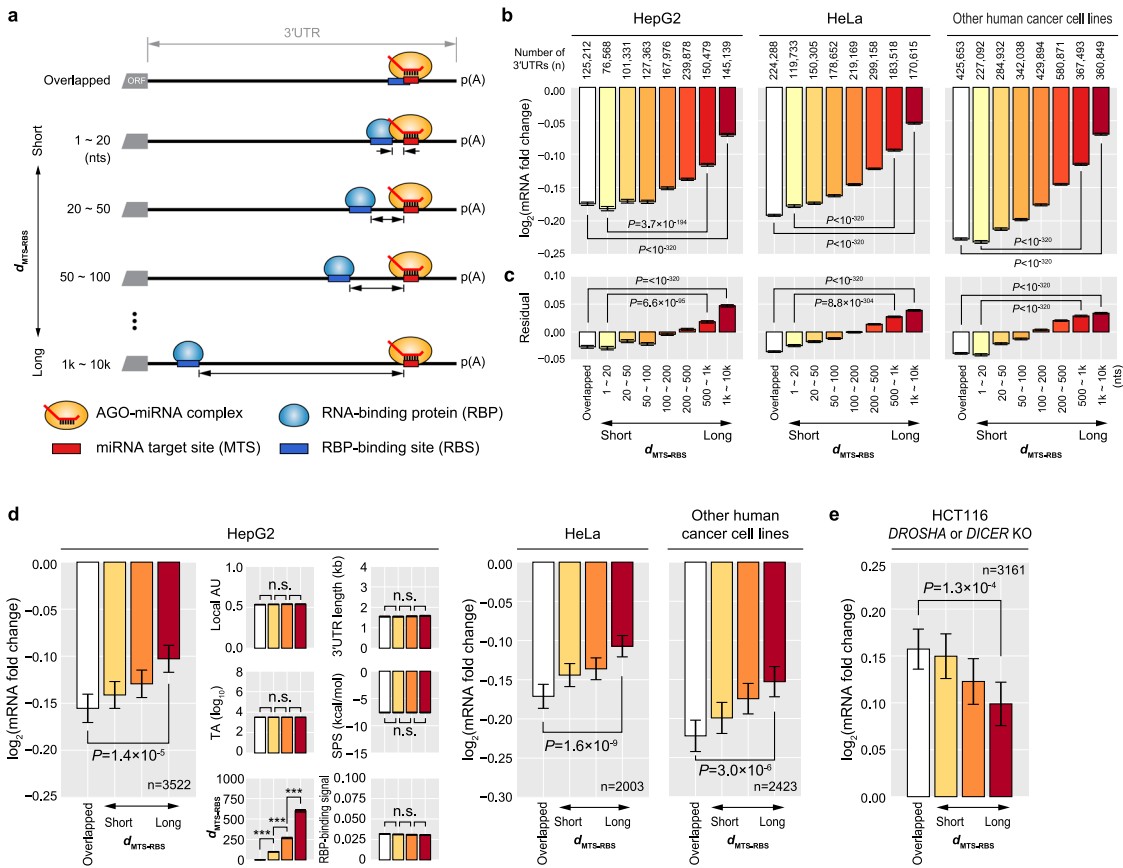

**Fig. 1 RBP binding close to a miRNA target site is associated with enhanced miRNA targeting. a** Overview of the analysis to investigate the effect of RBP binding on the miRNA targeting (MT) efficacy. Genes were grouped together based on the distance between a miRNA target site (MTS) and the nearest RBP-binding site (RBS) on the 3′UTR, denoted as $d_{MTS-RBS}$. The genes with an MTS that overlaps with an RBS were categorized as 'overlapped'. The association between MT efficacy and $d_{MTS-RBS}$ was analyzed by measuring the mRNA fold change. **b** Association analysis between $d_{MTS-RBS}$ and MT efficacy. The mean mRNA fold changes of 3′UTRs with a single 7, 8mer MTS obtained from mRNA-seq (HepG2) or microarray datasets (HeLa and other human cancer cell lines) that monitored the whole-transcriptome response to overexpressed miRNAs/siRNAs are shown. The 3′UTRs were grouped with respect to $d_{MTS-RBS}$, as depicted in (**a**). The mRNA fold changes were compared between the 3′UTR groups (two-sided Wilcoxon's rank-sum test). The number of 3′ UTRs used for measuring $d_{MTS-RBS}$ is shown on top. mRNA fold changes are displayed in the log2 scale and the error bars represent 95% confidence intervals. **c** Residuals of the mRNA fold change for each group from (**b**) after correcting for four known confounding features of MT (local AU content, target abundance (TA), seed-pairing stability (SPS), and 3′UTR length). The regression residuals represent the remaining information after reducing the contribution from the confounding features. Otherwise as in (**b**). **d** Association analysis between $d_{MTS-RBS}$ and MT efficacy after a more rigorous correction for the confounding features. A subset of 3′UTRs analyzed in (**b**) were selected and split into four subgroups with respect to the $d_{MTS-RBS}$. Each subgroup was carefully chosen to have statistically indistinguishable confounding features between each other (see Supplementary Fig. 1f for full versions). The mean values of confounding features, $d_{MTS-RBS}$, and log2(mRNA fold change) are displayed (***$P < 0.001$). Otherwise as in (**b**). **e** Association analysis between $d_{MTS-RBS}$ and MT efficacy after deleting *DROSHA* or *DICER*. Five miRNAs whose targets show the strongest derepression in response to miRNA removal were chosen and the association between $d_{MTS-RBS}$ and MT efficacy was evaluated. See Supplementary Fig. 1h for full versions. Otherwise as in (**d**).

Fig. 1b, in 100% of these RBPs that showed significant correlation, the shorter $d_{MTS-RBS}$ was associated with stronger MT efficacy indicating that all of these RBPs function as MT enhancers rather than suppressors; this is another striking inconsistency with the previously reported instances of RBPs that function as MT suppressors[10–12,36]. Taken together, these results demonstrate that for most RBPs, if not all, their binding close to the MTS is associated with enhanced MT efficacy, while no RBP is detectably associated with suppression of MT on a global scale.

To evaluate the collective effect of RBPs on MT, we investigated the association between MT efficacy and the number of bound RBPs in close proximity to MTSs. As a result, the number of RBPs was positively correlated with improved MT efficacy (Fig. 3a and Supplementary Fig. 2a, b), even when a small number of RBPs are bound near the MTS (Supplementary Fig. 2c). The correlation was also consistently observed for the MTSs with the preserved binding sites (Supplementary Fig. 2d)

and regardless of whether the RBP has specific binding motifs[37] or not (Supplementary Fig. 2e), suggesting that the observed impact of RBPs is quite general. To further examine whether the specific identities of RBPs instead of the number of bound RBPs determine MT efficacy, we have partitioned MTSs into two subgroups with similar numbers of the bound RBPs but with different identities of the RBPs (Fig. 3b, c and Supplementary Figs. 3, 4). Consequently, two subgroups did not show a significant difference in MT efficacy (Fig. 3c and Supplementary Figs. 3b, 4b, e). Therefore, instead of specific identities of RBPs, the overall number of bound RBPs to MTSs appears to be the primary factor that is associated with MT efficacy.

**A potential mechanism by which RBP binding enhances miRNA targeting.** One of the possible mechanisms that could explain the impact of RBPs on MT is protein–protein interaction between AGO and RBPs, as some RBPs have been previously

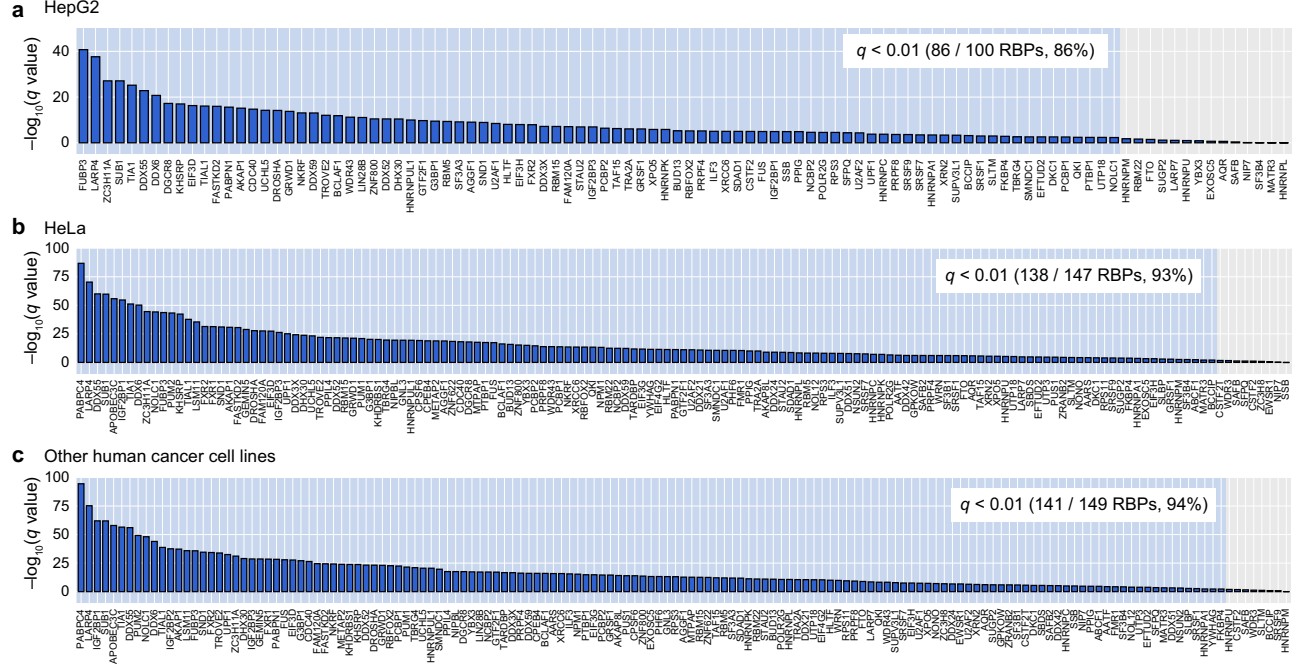

**Fig. 2 Most RBPs are associated with enhanced miRNA targeting. a** Association analysis of the distance between a miRNA target site (MTS) and the nearest RBP-binding site (RBS) on the 3′UTR, denoted as $d_{MTS-RBS}$ with miRNA targeting (MT) efficacy after correcting for four known confounding features (local AU content, target abundance, seed-pairing stability, and 3′UTR length) for individual RBPs. For each of the 100 RBPs whose RBSs in HepG2 cell line were identified in the ENCODE eCLIP-seq dataset, the association between $d_{MTS-RBS}$ and MT efficacy was tested after correcting for the four known confounding features for MT by the multiple linear regression (two-sided $t$-test). The $y$-axis represents the statistical significance of the observed association: upward and downward directions indicate the MT enhancement ($-\log_{10}(q$ value)) and the MT suppression ($\log_{10}(q$ value)) by the RBP, respectively. The number of RBPs that exhibits a significant association between $d_{MTS-RBS}$ and MT efficacy after multiple test correction by the false discovery rate is shown on top. **b**, **c** Association analysis between $d_{MTS-RBS}$ and MT efficacy by using 59 microarrays that measured the whole-transcriptome response to ectopically introduced miRNAs or siRNAs into HeLa cell line (**b**), and for 75 microarrays that measured the whole-transcriptome response to ectopically introduced miRNAs into various human cancer cell lines (**c**). Otherwise as in (**a**).

identified as direct interactors with miRISC[38,39] and have been revealed to regulate MT for several miRNA targets[9]. However, when partitioning RBPs into direct interactors with AGO and the others, both exhibited a significant correlation between $d_{MTS-RBS}$ and MT efficacy (Fig. 4a), indicating that both groups of RBPs may function as MT enhancers. Therefore, a more plausible mechanism to explain this general impact of RBPs on MT would be that RBP binding alters the local secondary structure of an MTS or its vicinity such that AGO can more readily access the MTS, in a similar manner that Pumilio and PCBP2 regulate the miRNA targets[6,7].

To examine whether RBP binding leads to the opening of the local secondary structures, we obtained a dataset of DMS-seq that detects unpaired adenines and cytosines at a nucleotide resolution enabling us to accurately probe in vivo secondary structures of endogenous RNAs[40]. When selecting three groups of 3′UTR fragments that have none, lenient, and stringent RBS signals, the 3′UTR fragments with lenient and stringent RBS signals had substantially unpaired secondary structures than those with none and lenient RBS signals, respectively (Fig. 4b). It is noteworthy that when selecting these three groups we have carefully chosen 3′UTR fragments that have statistically indistinguishable RNA secondary structure in vitro and mRNA expression levels among these groups. Thus, the elevated level of DMS score can be attributed to higher RBP-binding activities instead of RBPs already bound to the open structures.

This association between the RBP binding and elevated DMS-seq was more pronounced when a larger number of RBPs bind to the 3′UTR (Fig. 4c), consistent with our previous observation that the overall number of bound RBPs determines MT efficacy

(Fig. 3a). When looking into individual RBPs, 99% of RBPs exhibited significant association, indicating that RBP binding generally results in an opening of the local secondary structure of the 3′UTR in vivo (Fig. 4d), and this could be a common mechanism by which RBPs enhance MT efficacy. Additionally supporting this hypothesis, analysis of AGO2 PAR-CLIP-seq in response to ectopically introduced miRNA[18] exhibited significant increased AGO2 occupancy for those MTSs that have enriched RBP binding (Fig. 5a).

Another possible hypothesis that could explain the correlation between RBP binding and MT efficacy is AGO-mediated recruitment of RBPs where the MTSs of the transfected miRNAs promote the recruitment of RBPs. However, we examined the MTSs of the ectopically introduced miRNAs, which were apparently absent in the WT cells where eCLIP-seq was performed and thus unable to affect eCLIP-seq results. Therefore, the enrichment of RBP binding near the effective MTSs for the transfected miRNAs cannot be explained by the AGO-mediated recruitment of RBPs.

Therefore, we propose a model that takes RBP binding into consideration when explaining MT (Fig. 5b). Compared to the conventional model of ternary interaction among AGO, miRNA, and target mRNA, our proposed model better explains MT efficacy for multiple RBPs (Supplementary Fig. 5a). These results demonstrate that one of the main mechanisms by which RBPs influence MT may be that RBPs open up local secondary structures close to the MTS such that AGO can more easily access the MTS thus improving MT. However, our proposed mechanism does not rule out the previously reported mechanism where protein–protein interaction between AGO and some RBPs

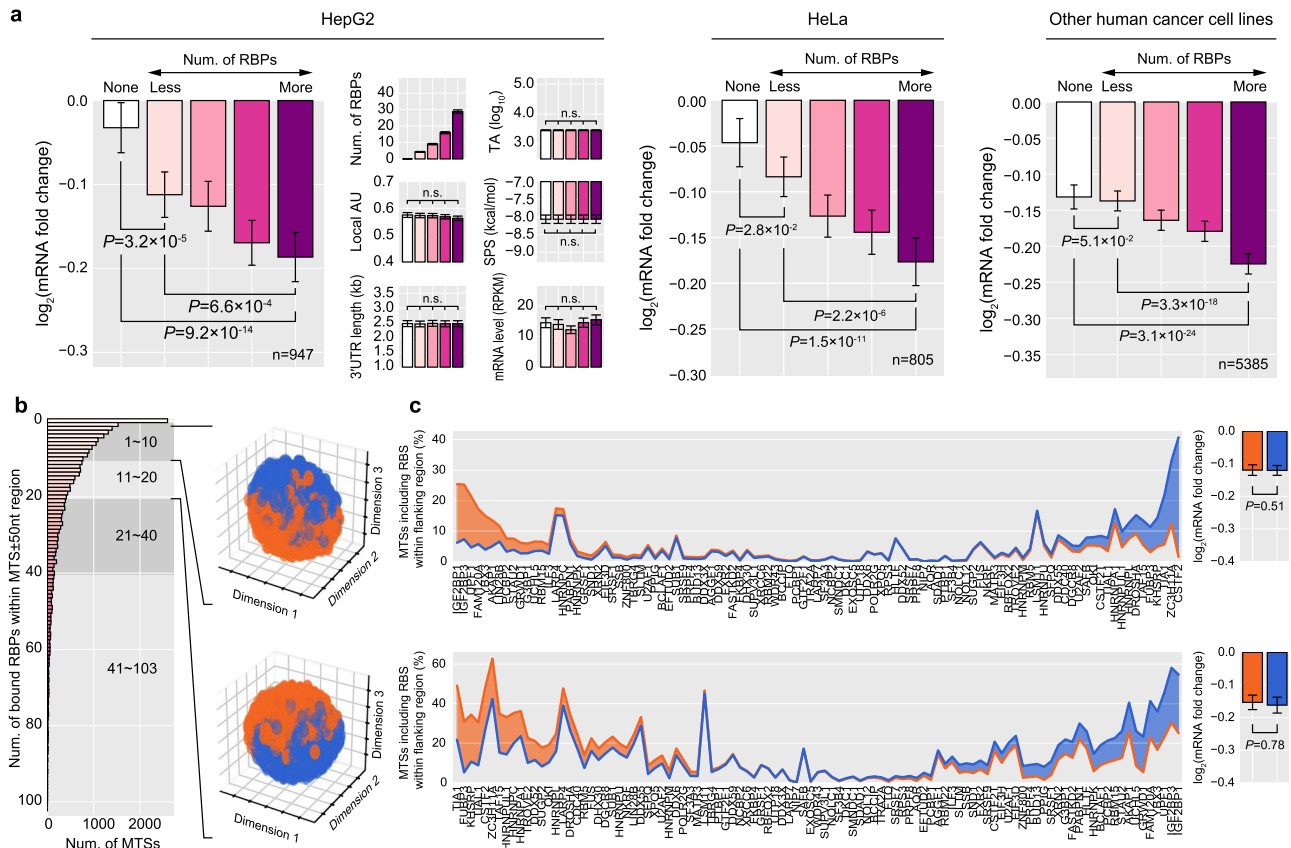

**Fig. 3 Combined effect of RBPs on miRNA targeting (MT). a** Association analysis between the number of RBPs bound close to the miRNA target site (MTS) and MT efficacy. 3′UTRs were separated into five subgroups with respect to the number of RBPs bound within 50 nts from the MTS. Each subgroup was carefully selected to have statistically indistinguishable confounding features among subgroups (see Methods). mRNA fold changes and values of confounding features were compared among these subgroups (two-sided Wilcoxon's rank-sum test) with the mean values of confounding features and $\log_2$(mRNA fold change) displayed. The error bars represent 95% confidence intervals of the values. For full versions for HeLa and other human cancer cell lines, see Supplementary Fig. 2a, b. **b** Distribution of MTSs with respect to the number of RBPs bound within 50 nt flanking regions in HepG2 data (left) and the separation of MTSs by the composition of bound RBPs (right). MTSs with the number of bound RBPs between 1 and 10 were collected and separated into two subgroups with similar number of bound RBPs but different RBP compositions (top right, see Methods). Similarly, those MTSs with the number of RBPs between 11 and 20 were collected and analyzed (bottom right). For full versions that also include the MTSs with the number of RBPs greater than 20, see Supplementary Fig. 3a. **c** For the two subgroups separated from the MTSs with the number of bound RBPs between 1 and 10, the RBP compositions (top left) and the mean MT efficacies (top right) are depicted and compared. For each RBP shown on the x-axis, the fraction of MTSs including an RBS within 50 nt flanking regions is displayed on the y-axis (top left). mRNA fold changes were compared between the two subgroups (top right, two-sided Wilcoxon's rank-sum test). Similarly, for the two subgroups separated from the MTSs with the number of bound RBPs between 11 and 20, the RBP compositions (bottom left) and the mean MT efficacies (bottom right) are depicted and compared. Otherwise as in (**b**). For full versions that also include the MTSs with the number of RBPs greater than 20, see Supplementary Fig. 3b.

mediates MT, and perhaps both mechanisms function together in a cooperative or independent manner depending on their cellular context.

**RBP binding generally enhances miRNA targeting**. To examine whether our proposed model is general enough to explain a wide spectrum of various RBPs, we partitioned RBPs into several subgroups with respect to mRNA stabilization function, RNA helicase activity, strand specificity, MTSs in 3′UTR or ORF, or by whether the RBP directly interacts with AGO (Figs. 4a, 6a–g, Supplementary Fig. 5b, c, and Supplementary Discussion). Accordingly, we observed a consistently significant association between $d_{MTS\text{-}RBS}$ and MT efficacy for all of subgroups. For instance, $d_{MTS\text{-}RBS}$ in ORF also exhibits a significant impact, albeit a modest degree (Supplementary Fig. 5c). Double-stranded RBPs and nuclear RBPs also seem to have enhancing effect on MT efficacy (Fig. 6e, f). To eliminate the concern of our results confounded by binding sites that overlap with those of single-

stranded RBPs or cytoplasmic RBPs, we only examined the RBPs with minimal overlapping binding sites based on the currently annotated data. Even when focusing on the subset of RBPs, the consistent results were observed (Supplementary Fig. 6 and Supplementary Discussion). Taken together, these results support that our findings are robustly general to be expanded to various RBPs and even to other regions such as ORF.

Next, we examined cases where an RBS overlaps with an MTS to inspect whether the competition between miRNA-loaded AGO and other RBPs hampers MT. Our previous results indicate that MT efficacy in such overlapping cases is greater than or equal to that of nonoverlapping cases (see white bar graphs in Fig. 1b–e), suggesting that miRNA-loaded AGO may easily outcompete RBPs. To more definitively investigate the potential competition, we have looked into various subset of MTSs separated by their site types, seed-pairing stabilities, and RBP functions including such cases where MT efficacy is expected to be very weak so that the potential competition between AGO and other RBPs gets

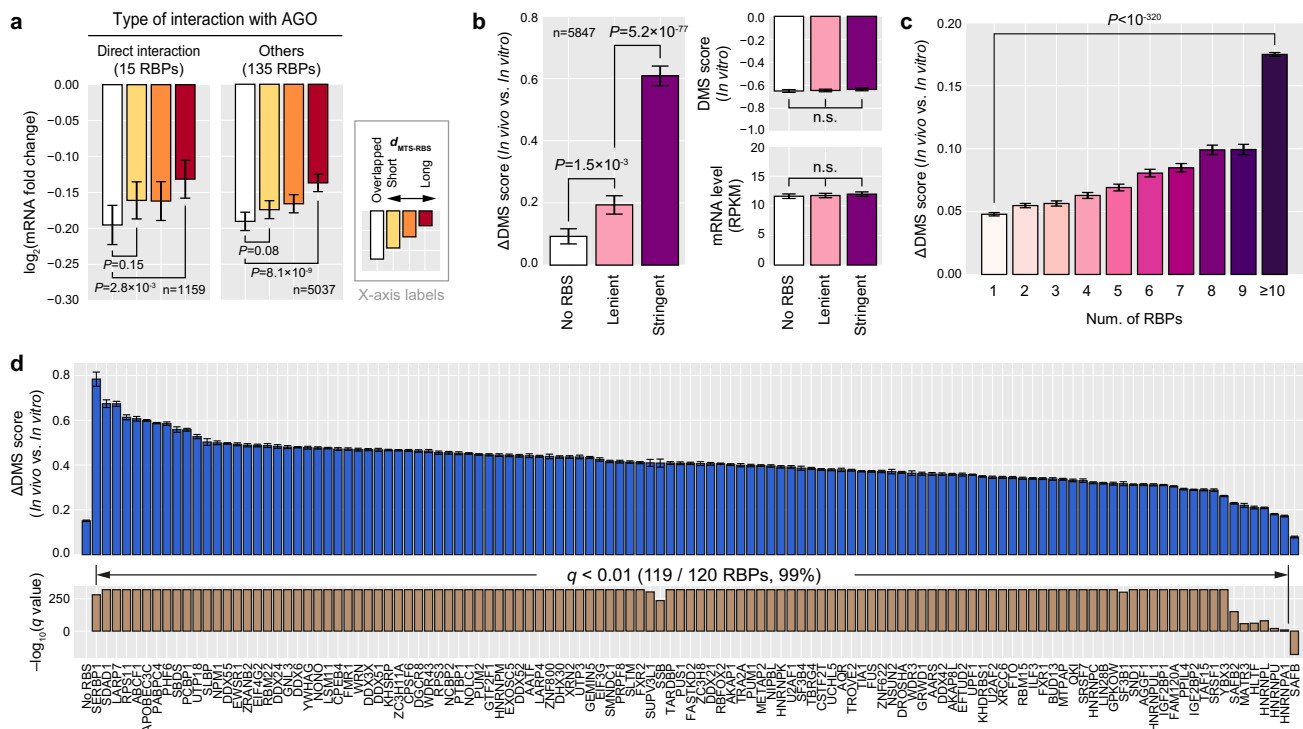

**Fig. 4 RBP binding opens up mRNA secondary structures. a** Consistent association between miRNA targeting (MT) efficacy and the distance between a miRNA target site (MTS) and the nearest RBP-binding site (RBS) on the 3′UTR, denoted as $d_{MTS-RBS}$, regardless of the direct interaction between RBP and AGO. 3′UTRs depicted in Fig. 1d were split into four subgroups with respect to the $d_{MTS-RBS}$ for RBPs directly interacting with AGO (left) or $d_{MTS-RBS}$ for the other RBPs (right). Each subgroup was carefully chosen to have statistically indistinguishable confounding features among four subgroups, and mRNA fold changes were compared among these subgroups (see Supplementary Fig. 5b for full versions). Otherwise as in Fig. 1d. **b** Association between RBSs and mRNA secondary structures. 100 nt fragments of human 3′UTRs were separated into three groups with respect to eCLIP-seq RBSs: 'No RBS', 'Lenient', and 'Stringent' include 100 nt fragments with none, leniently called, and stringently called eCLIP-seq RBSs, respectively. Each group was carefully selected to have statistically indistinguishable in vitro DMS scores and expression levels among three groups. For each of the 100 nt 3′UTR fragment, the change of DMS reactivity in vivo (ΔDMS score), which accurately represents the structural opening of RNA molecules in vivo[40], were calculated and compared among three groups (left, two-sided Wilcoxon's rank-sum test). Both eCLIP-seq and DMS-seq datasets monitored in K562 were used. The 'n' indicates the number of 3′UTR fragments. The mean ΔDMS scores are displayed and the error bars represent 95% confidence intervals. **c** Correlation between the number of RBPs and change of DMS reactivity in vivo (ΔDMS score) for 3′UTR fragments. ΔDMS scores of the first and last bins were compared (two-sided Wilcoxon's rank-sum test). Otherwise as in (**b**). **d** Change of DMS reactivity in vivo (ΔDMS score) for individual RBPs. 'No RBS' group consists of 3′UTR fragments without binding of any RBPs profiled in K562 cell line. For each group of fragments which contain specific binding sites of an RBP, ΔDMS scores were compared to that of 'No RBS' group (two-sided Wilcoxon's rank-sum test), and the statistical significance after multiple test correction by the false discovery rate is shown below. Otherwise as in (**b**).

more detectable. However, in all cases, MT efficacy was strongest for the MTS that overlaps with RBSs (Fig. 6a–g and Supplementary Fig. 5b), illustrating that miRNA-loaded AGO can outcompete RBPs instead of RBPs competing against AGO (Fig. 6h). Although this observed lack of competition between RBPs and miRNA-loaded AGO may seem counterintuitive at first, our observation is consistent with previous biochemical studies where the binding affinity for a single RBP and its target RNA is on average >1,000-fold weaker (dissociation constant $K_D$ in nanomolar range[41], Supplementary Table 1) than that for a miRNA and its mRNA target ($K_D$ in picomolar range[42], Supplementary Table 1). Our result is also consistent with a recently published study that some RBPs improve MT efficacy when the RBSs directly overlap with the MTS[43] (see Discussion).

**RBP binding enhances miRNA targeting by improving target-site accessibility.** To validate our proposed mechanism, we performed in vitro gel mobility-shift assays by using disrupted RBSs in three different 3′UTRs and recombinant RBPs (His-FUBP3 and His-PCBP2) (Fig. 7a, b and Supplementary Fig. 7a) and confirmed that the disruption of RBSs reduce RBP binding to mRNA targets (Fig. 7c). Successively, we performed gel mobility-

shift assays with recombinant human AGO2 protein and RBPs to confirm whether the disrupted RBSs also reduce AGO binding to the MTS. As a result, miRNA-loaded AGO2 bound to its mRNA target to a much weaker extent when the nearby RBS is disrupted (Fig. 7a, d), suggesting that RBP binding can improve the accessibility of MTSs to AGO2 in vitro. Since FUBP3 has been reported to directly interact with AGO[38,44] while PCBP2 has not been[45,46] (Supplementary Fig. 7b), our results suggest that RBPs generally enhance AGO binding to its MTSs regardless of whether they directly interact with AGO. We also confirmed that the accessibility of AGO is enhanced only when it is loaded with a targeting miRNA (Supplementary Fig. 7c) by the additional gel mobility-shift assay using AGO2 loaded with non-targeting miRNA (miR-1).

To monitor the transcriptome-wide structural change of RBSs upon RBP binding, we deleted *IGF2BP1* in HEK293T cell line (Supplementary Fig. 7d) and performed DMS-seq in parental and *IGF2BP1* KO cell lines. In parental cells, the stringent RBSs of IGF2BP1 exhibited greater DMS reactivity than that of lenient RBSs, demonstrating that the structure of these RBSs opens up upon binding of IGF2BP1 (Fig. 7e). Conversely, this pattern was reversed in IGF2BP1-depleted cells and DMS reactivity decreased

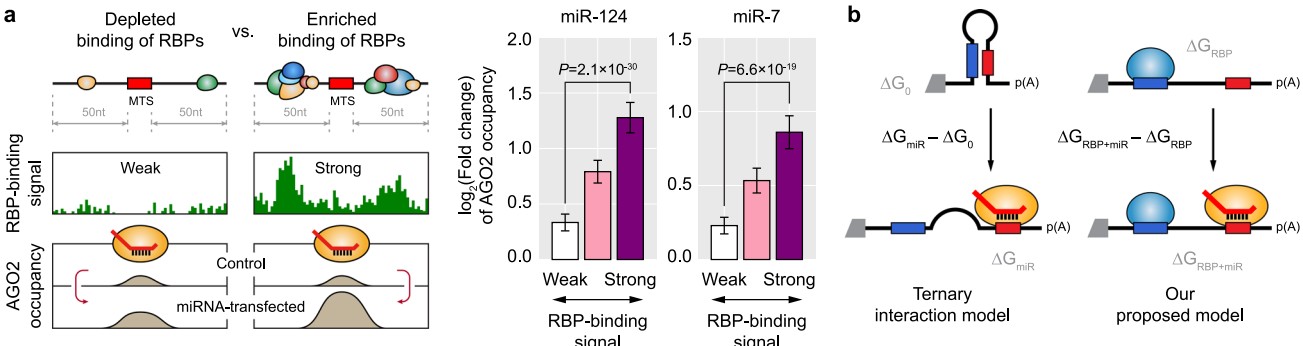

**Fig. 5 RBP binding enhances miRNA target-site accessibility of AGO. a** Association between enrichment of RBP binding and AGO2 occupancy. miRNA target sites (MTSs) of ectopically expressed miR-124 (middle) or miR-7 (right) were partitioned into three subgroups by the magnitude of overall RBP-binding signals within 50 nt flanking region of each MTS (left). Fold changes of AGO2 occupancy upon miRNA overexpression were calculated and compared among subgroups of MTSs (two-sided Wilcoxon's rank-sum test). The mean values of $\log_2$(fold change) of AGO2 occupancy are displayed, and the error bars represent 95% confidence intervals. **b** Proposed model that takes RBP-binding information into account when analyzing miRNA targeting in comparison to ternary interaction model among AGO, miRNA, and target mRNA. In the ternary interaction model, the changes between miRNA-unbound ($\Delta G_0$) and miRNA-bound ($\Delta G_{miR}$) MFEs are compared assuming there is no RBP binding in the 3′UTR. In our proposed model, additional information of RBP binding is incorporated and therefore the changes between miRNA-unbound ($\Delta G_{RBP}$) and miRNA-bound ($\Delta G_{RBP+miR}$) MFEs are expected to better reflect in vivo interactions between the RBS and MTS.

as the RBS signal stringency increased (Fig. 7e). The analysis indicates that when IGF2BP1 is absent, the secondary structure of the RBSs is in more closed state and this structural change is highly specific to the IGF2BP1-binding sites. We also tested whether RBP binding improves AGO binding to an MTS in vivo by AGO2-IP followed by RT-qPCR (Fig. 7f). Accordingly, the mRNA level bound to AGO2 in *IGF2BP1* knockout cells was significantly lower than that in the parental cells (Fig. 7g). When deleting another RBP, PCBP2 (Supplementary Fig. 7d), a consistent result was observed (Fig. 7g). Collectively, these results demonstrate that RBP binding improves target-site accessibility of MTSs to AGO both in vitro and in vivo.

To more directly evaluate whether RBP binding leads to improved MT in vivo, we performed luciferase reporter assays for ten 3′UTRs, each of which contains an MTS and a nearby RBS (Fig. 8a). When disrupting an MTS, nine of the ten 3′UTRs displayed a significantly reduced MT efficacy: among these nine 3′UTRs, eight exhibited significant reduction of MT in response to the disrupted RBS, showing that RBP binding indeed improves MT efficacy in vivo.

We further assessed the regulatory impact of RBPs on MT by using RBP knockout cells. When *PCBP2* is removed, all of six 3′ UTRs, each of which contains an MTS and a nearby PCBP2 RBS, exhibited reduced MT and then the reduced MT efficacy was restored after overexpressing PCBP2 protein (Fig. 8b). Similarly, in response to *IGF2BP1* deletion, both of the examined 3′UTRs, each of which contains an MTS and an IGF2BP1 RBS, exhibited reduced MT efficacy and then rescued when overexpressing IGF2BP1 protein (Fig. 8b).

To examine the impact of the RBPs on MT on a transcriptome-wide scale, we have performed mRNA-seq experiments after knocking out *PCBP2* or *IGF2BP1* and measured the whole-transcriptome response to overexpressed miRNAs. Accordingly, upon the deletion of IGF2BP1 or PCBP2, the miRNA targets were de-repressed for MTSs containing RBSs of the deleted RBP in the vicinity, while such derepression was not detected for non-targets or miRNA targets with MTSs located far from RBSs (Fig. 8c). KO of *IGF2BP1*, which had been previously reported as an MT suppressor, also led to the derepression of its target mRNAs (Supplementary Fig. 8a, b), indicating that although some RBPs can suppress MT of specific target mRNAs in a certain cellular context, more generally they function as MT enhancers on a

global transcriptome-wide scale. Taken together, our extensive validation experiments and analyses provide multiple solid lines of evidence that RBPs function as MT enhancers by improving the target-site accessibility to AGO.

**Evolutionary perspective on the regulatory impact of RBPs on miRNA targeting.** To gain a global insight into the impact of RBPs on MT, we investigated the locations of MTSs of 108 broadly conserved miRNA families[47] relative to RBSs on 3′UTRs (Fig. 9a). When considering RBPs profiled in HepG2 cell line, 90% of MTSs included ≥1 RBSs of these RBPs within 100 nt flanking regions. If focusing on conserved and highly conserved MTSs, this pattern was more prominent: 92% and 97% of conserved and highly conserved MTSs, respectively, had ≥1 RBSs within 100 nt flanking regions. Given that there are >1,500 human RBPs[15], our estimation may well be an underestimation and therefore we extrapolated our analysis to 1,500 human RBPs. Accordingly, we estimate that 100% of conserved and non-conserved MTSs have RBSs in their close proximity. When iterating our analysis for 120 RBPs profiled in K562 cell line, similar results were observed, implying that almost all of human MTSs are likely to be influenced by nearby RBSs.

To gain an evolutionary insight into MT and RBPs, for each of 54 human tissues[48], we examined whether RBSs of those RBPs expressed in a given tissue tend to co-occur or to mutually exclusively occur near MTSs of evolutionarily conserved miRNAs. As a result, significant enrichment of RBSs was observed near the MTSs in 33% of tested tissues, while significant depletion of RBSs was observed near the MTSs in none of tissues (Fig. 9b). This analysis indicates that locations of RBSs in 3′UTRs are evolutionarily selected to locate near MTSs perhaps to help enhance MT, providing an interesting perspective on MT and co-regulating RBPs. Based on these results, we propose a new revised model of MT that takes >1,500 co-regulating RBPs into account (Fig. 9c).

## Discussion

In our analysis for an RBS that overlaps with an MTS (Figs. 4a, and 6a–g), we suggest that binding affinity of miRNA-loaded AGO is much stronger (>1,000 fold) than that between an RBP and its mRNA target[41,42] and therefore miRNA-loaded AGO

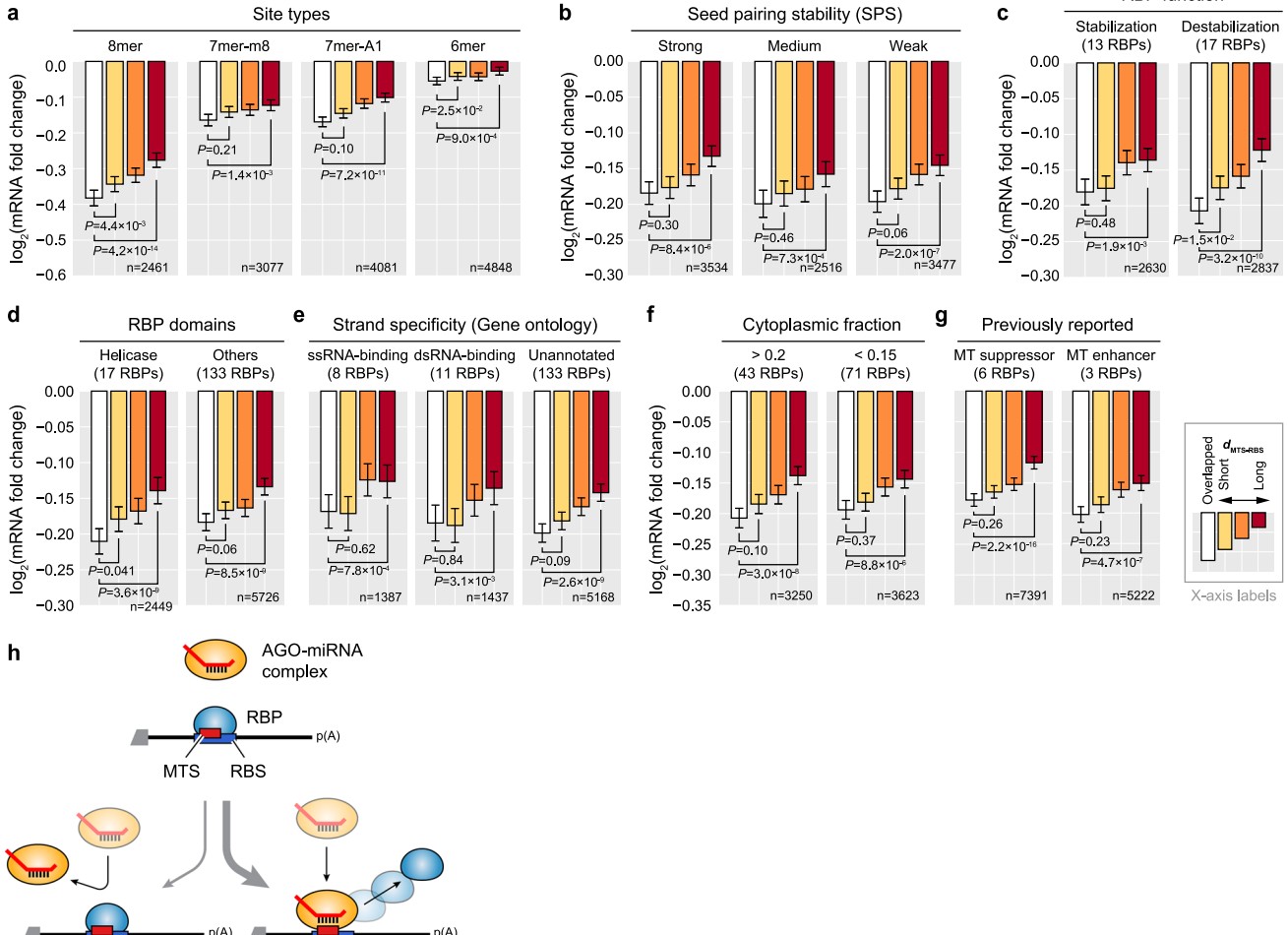

**Fig. 6 miRNA targeting is regulated by a wide range of RBPs. a–g** Association analysis of the distance between a miRNA target site (MTS) and the nearest RBP-binding site (RBS) on the 3'UTR, denoted as $d_{MTS-RBS}$, with MT efficacy for different site types, seed-pairing stabilities (SPSs), and various properties of RBPs. A subset of 3'UTRs depicted in Fig. 1d were selected with respect to different site types (**a**), SPSs (**b**), function of RBPs on mRNA stability (**c**), helicase activity (**d**), strand specificity (**e**), subcellular localization (**f**), and previously reported function of RBPs on MT regulation (**g**). The selected 3'UTRs were split into four subgroups with respect to $d_{MTS-RBS}$. Each subgroup was carefully chosen to have statistically indistinguishable confounding features among four subgroups, and mRNA fold changes were compared (see Supplementary Fig. 5b for full versions). Otherwise as in Fig. 1d. **h** Schematic illustration of the MT mechanism for the MTS that overlaps with an RBS. Before AGO-miRNA complex binds to the MTS, RBPs can bind to their RBS (top). One possible scenario afterwards is that the RBP binding competes against miRNA-bound AGO preventing it from productive MT (bottom left). However, our results support the other scenario where miRNA-bound AGO predominantly replaces mRNA-bound RBPs leading to productive MT (bottom right).

may easily outcompete the RBPs. If so, given that there exist numerous structured RNAs such as an RNA hairpin in 3' UTRs[49], how can the RBPs open the closed RNA structure in the first place? According to a previous transcriptome-wide study, mRNAs were observed to be less structured in vivo compared to in vitro, caused by combinatorial effects of both active mechanisms such as RNA helicase activity and passive mechanisms by binding of RBPs[40]. A large number of RBPs are reported to have helicase activity[50] and RBPs are considered to function together to accumulatively induce structural opening of RNAs[51]. For instance, an RBP with a helicase activity can initiate opening up a closed RNA structure and then other RBPs can be additionally recruited to induce further structural changes[52] (Supplementary Discussion). Consistent with this hypothesis, we observed that as the number of bound RBPs increases, the RNA structure tends to be more open (Fig. 4c). Once the RNA becomes less structured, miRNA-bound AGO readily accesses the MTS by replacing those RBPs already bound (Fig. 5a), eventually leading to productive MT (Fig. 3a).

Although the present study is based on rigorous bioinformatics analyses and experimental validations, it has the following limitations. First, the global analyses employed the whole-transcriptome datasets of microarray and mRNA-seq. This approach can be justified by the fact that mRNA destabilization rather than translational repression is the dominant mode of MT in steady state and therefore our approach is capturing most of relevant effects of MT[1,53,54]. However, to more conclusively address this issue, future efforts can be made to generate and analyze large-scale proteomics and/or ribosomal footprint datasets, attempting to dissect the contribution of RBPs to translational repression of MT compared to mRNA destabilization. Second, our study focuses on the role of RBPs in MT only in steady-state level and thus lacks an approach to investigating the dynamics of MT. As previously reported, the regulatory mode of MT in transient state can be different from that of steady state depending on the biological contexts[53,55]. Therefore, future studies that aim to revisit the role of RBPs in the context of dynamic regulation of MT such as the maternal to zygotic transition may

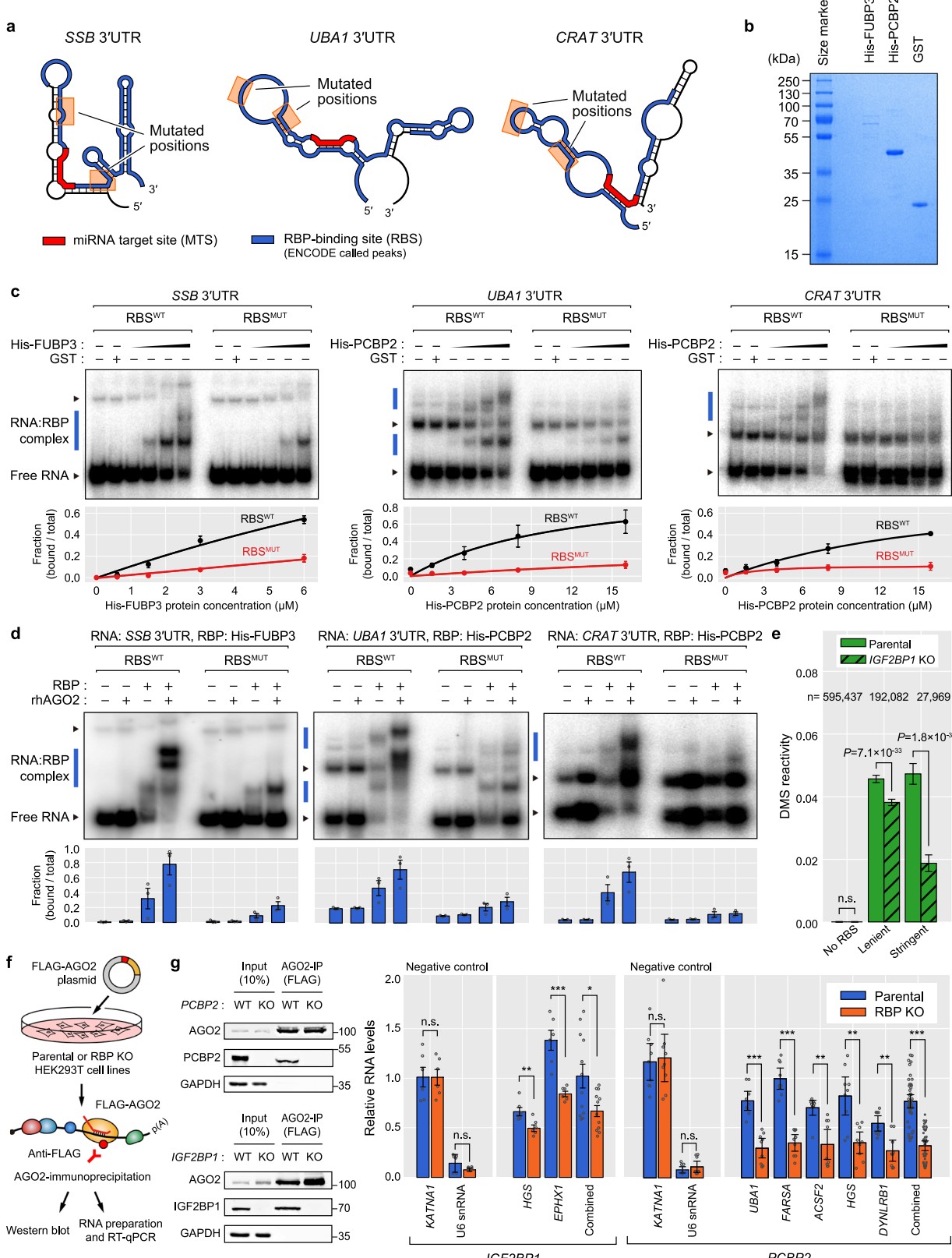

help draw a more complete picture of MT. Third, we employed a set of representative mRNA isoforms, which are not specific to a particular cell line throughout our analyses. Although use of mRNA annotation specific to the corresponding cell line can provide more accurate information on the existence of the target

sites, the overall association observed by our analyses is not largely dependent on cell-line-specific conditions. Our consistent results using the representative isoforms in various biological contexts support our claim of the general regulatory impact of RBP binding on MT efficacy. Since 3′UTR isoforms vary between

**Fig. 7 Disrupted RBSs or RBP knockout reduce the target-site accessibility to AGO. a** 3′UTR structures of RNA substrates (*SSB*, *UBA1*, and *CRAT*) used for gel mobility-shift assays shown in **b–d**. The 3′UTRs of SSB and UBA1 were also used as targets for luciferase assays in Fig. 8a, b. The miRNA target sites (MTSs) and RBP-binding sites (RBSs) are shown as red and blue boxes, respectively, with the mutated positions indicated by orange boxes. **b** Histidine-tagged recombinant RBPs (His-FUBP3 or His-PCBP2) and GST proteins were used for gel mobility-shift assays shown in (**c, d**). **c** Gel mobility-shift assays for RBS$^{WT}$ and RBS$^{MUT}$ with His-FUBP3 or His-PCBP2 with GST protein used as a negative control (top). The free RNA and RNA:RBP complex bands are shown as black and blue rectangles, respectively. Mean fractions of the bound RNA:RBP complexes ±95% confidence intervals are displayed (bottom, $n = 3$). **d** Gel mobility-shift assays with (1) 3′UTR, (2) 3′UTR and rhAGO2, (3) 3′UTR and RBP, and (4) 3′UTR, RBP, and rhAGO2. Otherwise as in (**c**). **e** DMS reactivities on A and C nucleotides of 3′UTRs were measured by comparing DMS counts of DMS(−) and DMS(+) samples. Corresponding nucleotides were divided into three groups by the RBP-binding signal of IGF2BP1 eCLIP-seq dataset (No RBS, Lenient, and Stringent). DMS reactivities between WT and RBP KO were compared (two-sided Wilcoxon's rank-sum test). The mean DMS reactivity values ±95% confidence intervals are displayed. **f** Experimental procedure for AGO2-immunoprecipitation (IP) followed by western blot and RT-qPCR. **g** AGO2-IP followed by western blot and RT-qPCR in AGO2-overexpressed HEK293T parental and RBP KO (*IGF2BP1* or *PCBP2*) cells. Protein levels in the input and IPed samples were visualized by the western blotting (left). Relative RNA levels of each input and IPed samples were quantitated in parental and RBP KO cells and normalized to each input sample (right, two-sided Wilcoxon's rank-sum test, \*$P < 0.05$, \*\*$P < 0.01$, \*\*\*$P < 0.001$, $n = 9$ for *PCBP2* and $n = 6$ for *IGF2BP1*). The mean relative RNA level of the IPed samples ±95% confidence intervals are displayed. The RNA level of *KATNA1* was used as a negative control and U6 snRNA-level served as a technical control of AGO2-IP. *P* values are provided in Source Data.

cells, these alternative isoforms can affect MT efficacy by inclusion or exclusion of MTSs between different cellular contexts[56]. Cell-type-specific MT regulatory mechanism by different 3′UTR isoforms is an interesting hypothesis worth investigating in the future. Fourth, although we reported that dsRBPs and nuclear RBPs also function as MT enhancers (Fig. 6e, f and Supplementary Fig. 6), the result should be still carefully interpreted because there are caveats of incomplete eCLIP-seq datasets and UV crosslinking bias to single-stranded regions[57]. These concerns may be alleviated when more comprehensive eCLIP-seq datasets and unbiased methods to capture RNA-RBP interactions become available, that would allow us to comprehensively study the effects of the RBPs on MT.

In this study, based on massive-scale analyses of binding sites for >100 human RBPs and on extensive validation experiments, we report that most RBPs enhance MT instead of suppressing it on a global scale, by making the local secondary structure of the MTS more readily accessible to AGO. Our study raises a challenging question about the broadly accepted model of MT that consists of a simple ternary interplay between AGO, miRNA, and mRNA target, proposing a largely revised model that takes a much broader context of >1,500 co-regulating RBPs into additional consideration (Fig. 9c). Our study illuminates the previously unappreciated regulatory impact of RBPs on MT, unraveling the complex nature of the gene regulatory network governed by metazoan miRNAs and their co-regulating RBPs. Undoubtedly, the RBP-binding information, if carefully combined with known determinants of MT, will help more accurately identify functional miRNA targets.

## Methods

**Cell culture**. HepG2 (ATCC, HB-8065) and HeLa cells (ATCC, CCL-2) were cultured in DMEM (Gibco) supplemented with 10% FBS (Gibco) and 1% Penicillin-Streptomycin (Gibco). Parental and RBP knockout (*PCBP* KO and *IGF2BP1* KO) of HEK293T cells (ATCC, CRL-3216) were cultured in same condition. HCT116 cells (ATCC, CCL-247) were cultured in McCoy's 5A media (Welgene) with 10% FBS (Gibco) and 1% Penicillin-Streptomycin (Gibco).

**Processing of RBP-binding information of eCLIP-seq data**. The ENCODE database provides both raw sequencing data and their processed data of the exact genomic location of RBP-binding sites (RBSs) detected from HepG2 and K562 cell lines[16]. Binding information of 103 RBPs in HepG2 cell line and 120 RBPs in K562 cell line, 150 RBPs in total, were analyzed. Since two replicates of binding information are provided for each RBP, the merged binding information of RBPs were used for analyses in corresponding cell line and thus the RBSs that have been detected only in one replicate was also included in our list of RBSs. When analyzing the data of other cell lines rather than HepG2 or K562, RBP-binding information of both cell lines was used. In case there exist data of RBPs profiled in both cell lines, RBSs were merged together.

To examine the impact of individual RBPs on miRNA targeting (MT) efficacy in Fig. 2a–c, following filtering steps were applied for selecting RBPs to be used in the analyses. Among the total profiled RBPs, SERBP1 was discarded since it has a low number of RBSs in the 3′UTRs (<300) while other RBPs have >1,000 RBSs in the 3′UTRs. When analyzing HepG2 and HeLa datasets, those RBPs whose expression levels belong to the lower 50% were additionally excluded. As a result, 100 out of 103 RBPs, 147 out of 150 RBPs, and 149 out of 150 RBPs were used for investigation on the datasets of HepG2, HeLa, and other human cancer cell lines, respectively.

When analyzing the association between the RBP binding and mRNA secondary structures (Fig. 4b–d), those RBSs that have strong statistical significance, which the ENCODE processed data provides, were defined as 'stringent RBS', while the others as 'lenient RBS'.

**Multiple linear regression analysis**. We used multiple linear regression (MLR) to correct for confounding factors known to influence MT (Figs. 1c, 2a–c, and Supplementary Fig. 5a). In order to correct for the effect of potentially confounding features contributing to MT (local AU content, target-site abundance, seed-pairing stability of corresponding miRNA, and 3′UTR length)[27–34], MLR models with aforementioned features were fitted to the log$_2$(fold change) of the target mRNAs for each dataset of HepG2, HeLa, and other human cancer cell lines. After fitting the MLR model, regression residuals were calculated by subtracting fitted values of the MLR model from the observed log$_2$(fold change) for each mRNA. The regression residual is interpreted as a remaining information of MT efficacy after correcting for the effect by four known confounding factors and it was used to test whether the distance between a miRNA target site (MTS) and the nearest RBS on the 3′UTR, denoted as $d_{MTS\text{-}RBS}$, is associated with MT efficacy (Fig. 1c). 'OLS.fit()' regression function in the 'statsmodels' package in Python[58] was utilized to fit the MLR models.

When investigating the association of RBP binding and MT efficacy for individual RBPs, MLR models were constructed for each RBP. From miRNA overexpression dataset of HepG2, HeLa and other human cancer cell lines, log$_2$(fold changes) for the mRNAs which contain a single 7, 8mer MTS on their 3′UTRs were collected, in order to clearly observe the impact of RBPs on a single MTS. For each RBP, the MLR model was fitted for the feature of RBP binding, either $d_{MTS\text{-}RBS}$ (Fig. 2a–c) or structural change of mRNA (Supplementary Fig. 5a), and the previously reported confounding features of MT. The association between MT efficacy and $d_{MTS\text{-}RBS}$ was measured by the *P* value of *t*-test. To correct the multiple testing problem, the *P* values were corrected by the false discovery rate (Fig. 2a–c).

**Correction for potentially confounding features of MT**. The confounding features of 3′UTR length, local AU content, TA, and SPS might also be potentially confounded with the RBP binding so need to be carefully controlled. To rigorously correct these confounding effects, we have devised an in-house algorithm facilitating to sample 3′UTRs into four different groups only according to $d_{MTS\text{-}RBS}$ while keeping the confounding effect similar across the groups. 3′UTRs containing a single 7, 8mer MTS and one or more RBSs were defined as the input of sampling process. The sampling process comprises of multiple rounds of selection. For a round of selection, a set of four 3′UTRs was selected with the following criteria: (i) In a same set, the site type of MTSs should be identical to one of the 8mer, 7mer-m8, and 7mer-A1, (ii) The min–max range of confounding effect should fall within the specified cutoffs, that were determined by the Gaussian process implemented in the 'bayes_opt' Python package to maximize the number of selected 3′UTRs, (iii) One 3′UTR should have an overlapping MTS with the nearest RBS ($d_{MTS\text{-}RBS} = 0$) and other three should have MTS separated from the nearest RBS ($d_{MTS\text{-}RBS} > 0$).

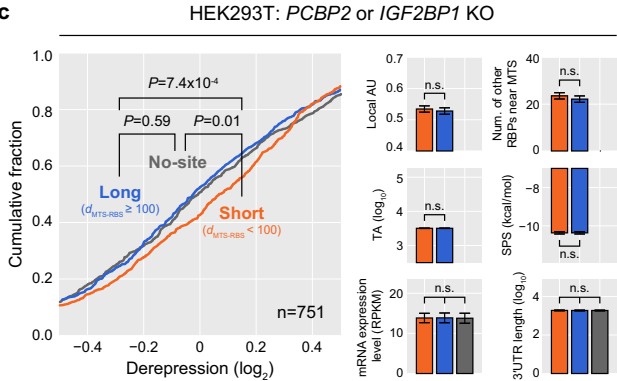

3′UTRs which met the criteria above were then assigned to the first to fourth groups in ascending order of the value of $d_{MTS-RBS}$. Next rounds of selection of four 3′UTRs proceeded until the inputs were exhausted. If any confounding factor exhibited significant difference across the groups, the sampling process was retried with the narrower cutoffs mentioned at the second criterion. After the sampling process finished, comparison of the log₂(fold change) or the confounding features across the groups was performed using Wilcoxon's rank-sum test of 'SciPy' package

in Python. Accordingly, we can clearly divide 3′UTRs solely dependent on $d_{MTS-RBS}$ without any concern that the confounding effect involves across groups. By using the dataset of *DICER* or *DROSHA* knockout in HCT116 cell lines[35], we performed the similar analysis to detect the derepression of target mRNAs. To select miRNAs whose targets show the strongest derepression in response to miRNA removal, a 2 × 2 contingency table was constructed for each miRNA by examining whether its MTS was included in the 3′UTR and whether the 3′UTR was highly de-repressed.

**Fig. 8 Experimental validation for the regulatory impact of RBPs on microRNA targeting. a** Fold repression of 3′UTRs with a miRNA target site (MTS) near an RBP-binding site (RBS) measured by luciferase reporter assay with the design of constructs shown left. Four different designed constructs were used: (1) wild-type MTS and RBS (MTS$^{WT}$RBS$^{WT}$), (2) mutated MTS (MTS$^{MUT}$RBS$^{WT}$), (3) mutated RBS (MTS$^{WT}$RBS$^{MUT}$), and (4) mutated MTS and RBS (MTS$^{MUT}$RBS$^{MUT}$). The normalized fold repressions were compared between MTS$^{WT}$RBS$^{WT}$ and MTS$^{WT}$RBS$^{MUT}$ (two-sided Wilcoxon's rank-sum test, $*P < 0.05$, $**P < 0.01$, $***P < 0.001$, $n = 12$, See Methods for normalization procedures). The changes in the minimum free energy of the 3′UTRs ($\Delta G_{miR} - \Delta G_0$ and $\Delta G_{RBP+miR} - \Delta G_{RBP}$) are listed at the bottom. *KATNA1* was used as a technical control. The mean values ±95% confidence intervals are displayed. *P* values are provided in Source Data. **b** The design of constructs and the expected changes in the secondary structures of mRNAs are shown (top left): parental, RBP KO, and rescue conditions of the deleted *RBP* are depicted by blue, orange, and green colors, respectively, with striped colors representing the mutated MTS. The protein abundance was quantified by western blot (top right). For the rescue experiment of each RBP KO, FLAG-tagged RBP was quantified (top right). GAPDH levels serve as a loading control. The MTS$^{WT}$ values were normalized to MTS$^{MUT}$ and were compared between parental and KO and between KO and rescue (two-sided Wilcoxon's rank-sum test, $*P < 0.05$, $**P < 0.01$, $***P < 0.001$, $n = 12$). Otherwise as in (**a**). **c** Transcriptome-wide response of miRNA targets after RBP removal. mRNA expression fold changes after miRNA overexpression were measured for *IGF2BP1* or *PCBP2* KO cells and compared to HEK293T parental cells, with the x-axis indicating the difference of the log$_2$(mRNA fold change) values. Target 3′UTRs with a single 7, 8mer MTS were selected if the distance between an MTS and the nearest RBS, denoted as $d_{MTS-RBS}$, is short (<100, orange) or long (≥100, blue). After controlling for confounding features of MT (right), distributions of log$_2$(mRNA fold change) were compared between the subgroups (left, two-sided Kolmogorov–Smirnov test). Distribution of miRNA non-targets ('No-site', gray) was plotted for comparison. The mean values of confounding features are displayed and the error bars represent 95% confidence intervals.

The most significant five miRNAs after $\chi^2$ tests were chosen and used for the analysis.

**DMS-seq analysis.** To distinguish whether RBP binding leads to the 3′UTR structural change, we used a DMS-seq dataset generated in K562 cell line with treatment of DMS (Accession ID: GSM1297493)[40] and the binding information of 90 RBPs from the ENCODE eCLIP-seq in K562 cell line. We monitored mRNA expression profile in K562 cells[23] and used top 50% of the most highly expressed mRNAs for the analyses because mRNAs with low expression level tend to have depleted signal of DMS-seq, which can lead to an inaccurate detection of unpaired regions for the mRNAs. Across the highly expressed human mRNAs, we collected 3′UTR fragments with all possible 100 nt windows shifted by every 10 nts. The 3′UTR fragments were separated into three groups: fragments that contain one or more RBSs with high confidence as explained above (stringent RBSs), fragments that contain RBSs with moderate signal only (lenient RBSs), and the other fragments without any RBS (No RBS). 3′UTR fragments were subsampled to have similar values of the predicted minimum free energy and the expression levels of mRNAs where the fragments are originated among the groups.

We normalized DMS read counts to RPM scale for each replicate of DMS-seq data. The DMS levels were then calculated by averaging the normalized counts of multiple replicates. We removed the background signal by comparing DMS levels between in vivo and denatured control. The resulting value, 'DMS score', is defined by Eq. (1). Also for in vitro sample, the DMS score was calculated by Eq. (2). For each selected fragment, ΔDMS score was measured by Eq. (3) where comparing DMS scores between in vivo and in vitro and the scores were compared among the groups (Fig. 4b). To assess the correlation between the number of bound RBPs and opening of the secondary structure of mRNAs, 3′UTR sequences with stringent RBSs were collected. These sequences were separated with respect to the number of bound RBPs, and their ΔDMS scores were normalized by the expression level of 3′UTR (Fig. 4c). The enrichment of ΔDMS score was tested by comparing the corresponding score in the region without any RBP binding (Fig. 4d). All statistical tests were performed by Wilcoxon's rank-sum test using 'SciPy' package in Python.

$$DMS\ score_{In\ vivo} = \log_2\left(\frac{DMS\ level_{In\ vivo}}{DMS\ level_{Denatured}}\right) \quad (1)$$

$$DMS\ score_{In\ vitro} = \log_2\left(\frac{DMS\ level_{In\ vitro}}{DMS\ level_{Denatured}}\right) \quad (2)$$

$$\triangle DMS\ score = DMS\ score_{In\ vivo} - DMS\ score_{In\ vitro} \quad (3)$$

The HEK293T parental and IGF2BP1 KO DMS-seq libraries were prepared using the protocol a previous study[40] with following modifications. The fragmented libraries were ligated to 3′ adapter from a recent study[59] with T4 RNA ligase2 truncated K227Q (NEB), and size-selected on a 10% UREA polyacrylamide gel. Next, the library was reverse transcribed and ligated to 5′ adapter[59] with RNA ligase 1 high concentration (NEB) at 22 °C overnight, as described in another previous study[16]. Then, the library was size-selected on a 10% UREA polyacrylamide gel and amplified by PCR. The final size-selected DMS-seq libraries were sequenced using HiSeq 2500 (single-end, 51 bp and 101 bp, Illumina). All the information of the primers and adaptors used in this study are listed in Supplementary Table 4. DMS count was calculated by detecting 5′ end of mapped reads (RT-stops). DMS reactivity was measured on A and C nucleotides of 3′UTRs by comparing DMS counts of DMS-treated and DMS-untreated samples (Fig. 7e).

**Gel mobility-shift assay.** To assess if RBPs generally enhance AGO binding to MTSs regardless of whether they directly interact with AGO, FUBP3, and PCBP2 were chosen for our gel mobility-shift assay because FUBP3 is reported to interact with AGO2[38] while PCBP2 has not been reported to interact with AGO2[45]. As one of FUBP3 binding targets, *SSB* 3′UTR, whose RBP-binding signal is indicated in Supplementary Fig. 7a, was chosen for the gel mobility-shift assay (Fig. 7c, d). To confirm a non-direct interactor enhances AGO2 binding to the 3′UTR, we have performed gel mobility-shift assays with *UBA1* and *CRAT* 3′UTRs that contain PCBP2-binding sites (Fig. 7c, d).

PCR fragments for 3′UTRs of *SSB*, *UBA1*, and *CRAT* containing T7 promoter were amplified from psiCHECK-2-*SSB*, -*UBA1*, and -*CRAT* plasmids, respectively. To prepare for RBS$^{MUT}$ sequences that we designed as described in Supplementary Table 4, we performed site-directed mutagenesis from the plasmids containing RBS$^{WT}$. The RNA substrates were synthesized with [$^{32}$P]-UTP by in vitro transcription and purified by resolving on a denature-PAGE. To prepare N-terminal Histidine-tagged protein, coding sequences of each *FUBP3* and *PCBP2* were amplified by specific primers (Supplementary Table 4) and inserted to pET-28b vector. Both recombinant histidine-tagged FUBP3 and PCBP2 proteins were purified by the HisTrap column (GE healthcare), according to manufacturer's instructions. The GST protein was purified by the GST affinity beads (Elpis Biotech) to be used as a control protein. The purified recombinant proteins were resolved on 12% SDS-PAGE and stained with Coomassie brilliant blue (BIORAD) (Fig. 7b).

Gel mobility-shift assays were performed as previously described[60]. Briefly, the reactions were carried out with the following condition: binding buffer (20 mM HEPES at pH 7.4, 50 mM KCl, 0.5 mM EDTA, and 10% (v/v) Glycerol), 1 mM DTT, radiolabeled 3′UTRs of *SSB*, *UBA1*, or *CRAT*, and gradient concentrations of recombinant proteins (His-FUBP3: 0.75, 1.5, 3, and 6 μM; His-PCBP2: 2, 4, 8, and 16 μM). The reaction was incubated at 37 °C for 10 min. Both free RNAs (unbound fraction) and RNA-RBP complexes (bound fraction) were resolved on 6% native-PAGE and analyzed by Typhoon FLA 7000 (GE healthcare). The assays were performed in triplicate and the average ratio between bound and total fractions was obtained by quantitating the band intensities using the Multi-gauge v3.0 program (Fig. 7c).

Recombinant human AGO2 (rhAGO2) was prepared using a Baculovirus expression system (Invitrogen) as previously described in detail[61]. Native gel mobility-shift assays with rhAGO2 proteins were performed in the following conditions: binding buffer, DTT, radiolabeled *SSB*, *UBA1*, or *CRAT* RNA substrates, recombinant proteins (His-FUBP3: 3 μM; His-PCBP2: 8 μM), and miRNA-loaded AGO2 proteins (*SSB*: miR-216a-5p; *UBA1*: miR-34a-5p; *CRAT*: miR-24-3p). For Supplementary Fig. 7c, non-targeting miRNA (miRNA-1) was loaded to rhAGO2 proteins. The reactions were resolved on 6% native-PAGE and analyzed by Typhoon FLA 7000 (GE healthcare). The assays were performed in triplicate and the average ratio between bound and total fractions was obtained by quantitating the band intensities using the Multi-gauge v3.0 program (Fig. 7d).

**RNA immunoprecipitation (IP) and qPCR.** FLAG-tagged human *AGO2* was cloned to pcDNA3.1 vector. HEK293T parental and RBP KO (*PCBP2* and *IGF2BP1*) cells were seeded in a six-well plate at ~60% confluency. After 24 h of incubation, the FLAG-hAGO2 plasmids were transfected into cells using Lipo-fectamine 2000 according to the manufacturer's protocol. After 24 h incubation, the cells were harvested, and then lysed by lysis buffer (20 mM HEPES-KOH at pH 7.4, 150 mM KOAc, 1.5 mM Mg(OAc)$_2$, 0.1% Triton X-100) with 1× Protease inhibitor cocktail (Roche). The precleared lysate was incubated with anti-FLAG M2 affinity gel (Sigma) at 4 °C overnight. Immunoprecipitated samples were washed four times with wash buffer containing 300 mM KOAc. Total RNAs were extracted by TRIzol (GeneAll) to be subjected to RT-qPCR.

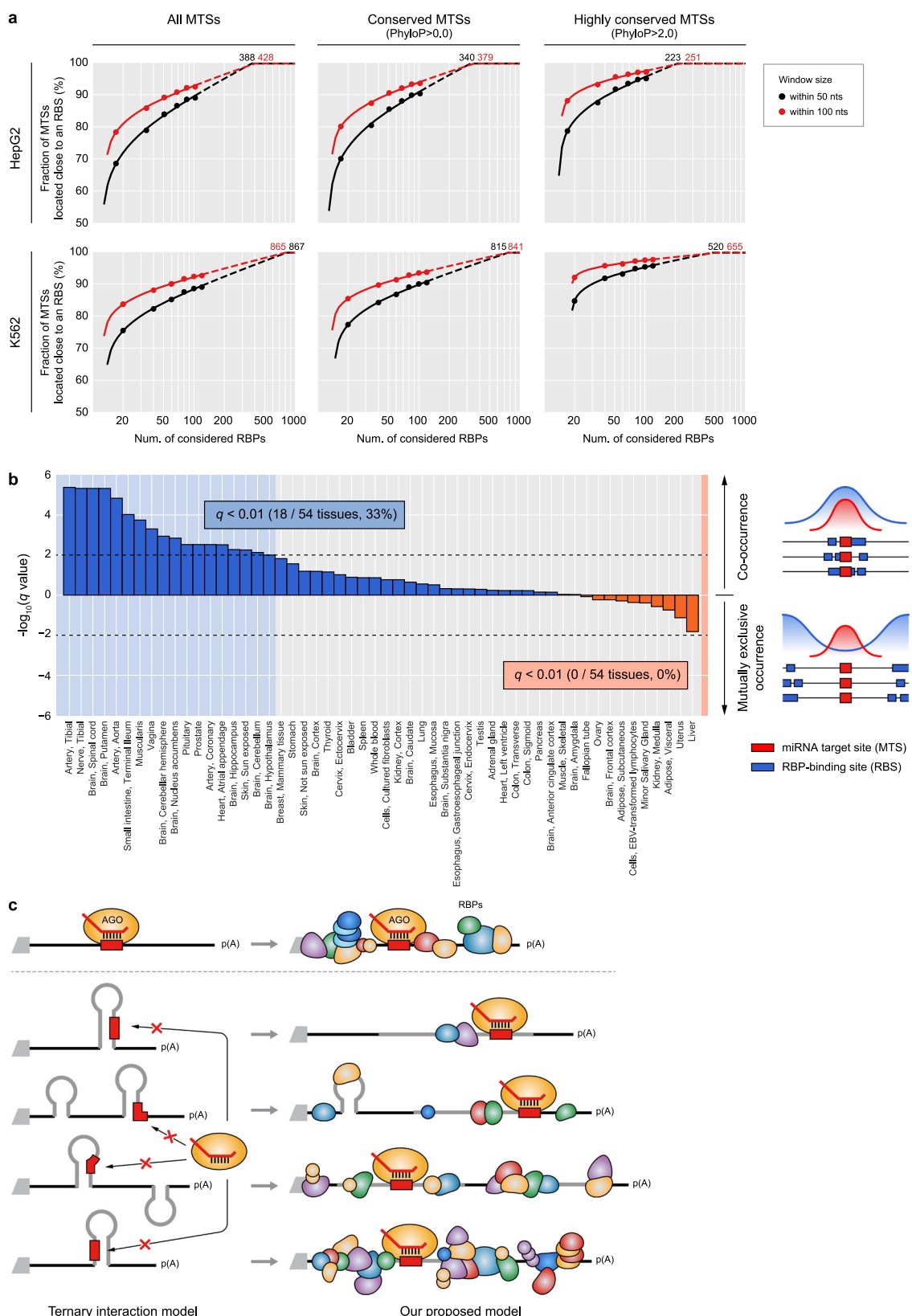

RNAs were isolated, treated with DNase I (Takara), and then converted to cDNA by reverse transcription using oligo dT primers and PrimeScript reverse transcriptase (Takara). Quantitative PCR was performed using SYBR green I master mix (Roche) with the gene-specific primers (Supplementary Table 4). As a technical control of the IP experiment, U6 snRNA was detected using Taqman miRNA assay (Applied Biosystem). The qPCR was performed with LightCycler II 480 v.1.5.1 (Roche) following the manufacturer's protocol. The Ct values of IP samples were normalized to those of each input sample using the following equations.

$$\triangle Ct_{\text{Input}} = Ct_{\text{Input}} - Ct_{\text{Input}} = 0 \qquad (4)$$

**Fig. 9 Evolutionary perspective on the regulatory impact of RBPs on miRNA targeting. a,** Fraction of miRNA target sites (MTSs) that have RBP-binding sites (RBSs) in their close proximity. 7, 8mer MTSs of miRNA families conserved across the vertebrates were collected[47]. The fraction of MTSs including ≥1 RBSs within 50 or 100 nt flanking regions was calculated using the binding information of RBPs profiled in HepG2 (top left). Based on the calculated fractions for subsampled RBPs (solid dots), we fitted a regression curve (solid line) to extrapolate the analysis to a larger number of considered RBPs (dashed line). The same analysis was repeated for conserved (PhyloP[62] > 0.0, top middle) and highly conserved MTSs (PhyloP > 2.0, top right). Similar analyses were performed for RBPs profiled in K562 (bottom). **b** Co-occurrence of evolutionarily conserved MTSs and RBSs in various human tissues. By utilizing transcriptome data of 54 tissues[48], highly expressed RBPs and miRNAs were collected for each tissue. For each conserved miRNA highly expressed in a given cell, 7, 8mer MTSs were collected and RBP-binding signals near the MTS were measured within 50 nt flanking regions. The magnitudes of RBP-binding signal were compared between the chosen MTSs and their control MTSs by using two-sided Wilcoxon's rank-sum test after correction of confounding features (Supplementary Fig. 8c). Statistical significances after multiple test correction via the false discovery rate are displayed. Upward and downward directions of the y-axis indicate co-occurrence and mutually exclusive occurrence between MTSs and RBSs, respectively.
**c** Proposed model of MT that takes >1,500 RBPs into account. For an MTS readily accessible to AGO, other RBPs would not affect MT efficacy (top row) and such cases are well explained by the currently accepted model of a ternary interplay between AGO, miRNA, and mRNA target. However, as our analysis indicated that in a vast majority of cases ≥1 RBSs are located close to the MTS, our proposed model that additionally takes RBSs of >1,500 3'UTR-binding RBPs into consideration would better explain MT (bottom four rows).

$$\triangle Ct_{\text{IPed}} = Ct_{\text{IPed}} - Ct_{\text{Input}} \qquad (5)$$

The relative RNA levels of IPed samples ($2^{-\triangle Ct_{\text{IPed}}}$) are shown in the figure. The IP experiments were performed in nine and six replicates with *PCBP2* KO and *IGF2BP1* KO cells, respectively (Fig. 7g).

**Luciferase reporter assays**. Luciferase reporter assays were carried out as previously described in detail[32]. Briefly, HepG2 cells were seeded in 96-well white plates (Greiner Bio-one) and then co-transfected with 50 ng of psiCHECK-2 reporter plasmids containing wild-type or mutated constructs for each candidate and 75 nM of its cognate miRNA duplexes using Lipofectamine 3000 (Invitrogen). The 3'UTRs used in the luciferase reporter assays were selected based on criteria as described above. The primers are listed in Supplementary Table 4. After 48 h, the luciferase activities were estimated by the Dual-Glo luciferase assay (Promega) and GloMax 96 Microplate Luminometer v.1.9.2, according to the manufacturer's protocol (Fig. 8a).

HEK293T parental and knockout cell lines (*IGF2BP1* and *PCBP2*) were cultured in DMEM supplemented with 10% FBS (Gibco), seeded at ~60% confluency, and co-transfected with reporter plasmids and miRNA duplexes using Lipofectamine 2000 (Invitrogen). The cells were incubated for 24 h and then its luciferase activities were measured by the Dual-Glo luciferase assay. Firefly luciferase activity was normalized to *Renilla* luciferase activity. For each sample, 12 replicates of assays were performed (Fig. 8b).

**Reporting summary**. Further information on research design is available in the Nature Research Reporting Summary linked to this article.

## Data availability
The data supporting the findings of this study are available from the corresponding authors upon reasonable request. The raw sequencing data, expression levels, and fold changes used in this study are available in the Gene Expression Omnibus database under accession number GSE115646. Source data used for creating all figures are provided as a Source Data file with this paper. Source data are provided with this paper.

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

## Acknowledgements

This study was supported by the National Research Foundation of Korea (NRF) funded by the Ministry of Science and ICT, Republic of Korea (NRF-2014M3C9A3063541, NRF-2019M3E5D3073104, NRF-2020R1A2C3007032, and NRF-2020R1A5A1018081), Korea Health Industry Development Institute (KHIDI) funded by the Ministry of Health and Welfare, Republic of Korea (HI15C3224) and the Institute for Basic Science (IBS) from the Ministry of Science and ICT of Korea (IBS-R008-D1) to D.B. This work was also supported by Cooperative Research Program for Agriculture Science and Technology Development (Project No. PJ01577601) Rural Development Administration, Republic of Korea and the NRF grant funded by the Ministry of Science and ICT, Republic of Korea (NRF-2021R1A5A1032428) to C.S.

## Author contributions

D.B. and J.-W.N. conceptualized the study; Y.K.K., C.S. and D.B. developed the methodology; Su.K. and D.K. designed and performed bioinformatics analysis; So.K. and H.R.C. designed and performed experimental validation; Su.K., So.K., H.R.C., D.K., Ju.P., N.S., Jo.P., M.Y. and G.C. analyzed data and interpreted results; Y.-K.K., V.N.K., C.S. and D.B. provided resources for this study; Su.K., So.K., H.R.C., D.K. and D.B. wrote the manuscript with comments from all authors; C.S. and D.B. supervised this study.

## Competing interests

The authors declare no competing interests.
