## [Peer Review File · Nature Communications]

REVIEWER COMMENTS

Reviewer #1 (Remarks to the Author):

In this manuscript, Kim et al., examined the impact of RBPs on the efficiency of miRNA targeting. They leveraged a large collection of eCLIP datasets from ENCODE to identify binding sites of 117 RBPs, and correlated their binding (distance and abundance) with the efficiency of miRNA mediated gene repression. They generally observed RBP binding promoting miRNA repression. They provided individual examples for the enhancement of RBP binding to the binding of the RISC in vitro. They also knocked out two RBPs, PCBP2 and IGFBP1, and showed Ago2 associated mRNAs were reduced in both cases. Overall, this is a potentially interesting study, leveraging the power of large-scale genomic datasets and bioinformatic analysis to study an important question in RNA biology and gene expression regulation. However, I have several major concerns that should be carefully addressed by the authors.

First, the main conclusion, RBPs globally enhance miRNA targeting, is largely based on extrapolation. Specifically, they used 117 RBP eCLIP datasets to generate the global RBP-binding sites (RBS) information, which is critical to calculate the distance between miRNA binding sites and RBS. However, 117 RBPs are less than 10% of 1,500 RBPs, and it was unknown how many of these 1,500 RBPs are expressed in HepG2 and HeLa cells, the cell source of their eCLIP data. Even with a conservative estimation, 30% of the total RBPs should be expressed at a reasonable level in these cells, which suggest ~450 RBPs can bind to mRNAs, far more than 117 RBPs. Without such knowledge, it is impossible to accurately determine the true distance between any RBS to MTs (miRNA target sites) because any unbound mRNA sequences could be occupied by undetermined RBPs. In addition, they have not further narrowed down the RBS by searching for any specific binding motif for any RBPs. Without the binding motif information, the definition of RBS is not strong. Both of these two issues are problematic for the relatively long dMTS-RBS cases such as 500~1k or 1k~10k categories. They could be, in fact, much shorter distance to nearby MTS. Yet, they still observed weaker effect on miRNA repression of these apparently long dMTS-RBS sites.

Another critical component of this analysis is miRNA target sites (MTS). They have not clearly stated how they identify MTS. Based on the Method section (pg. 54), they selected 28 conserved but not very highly expressed miRNA for overexpression. Then they just used 8mer, 7mer-m8, 7mer-A1 and 6mer, presumably the different matches to the seed sequences of these miRNAs, to define MTS (pg. 54, line 871), which is clearly not adequate given the high false positive rates of identifying MTS by the seed match alone. Next, they heavily selected a portion of mRNAs for their analysis – they discarded upregulated genes upon miRNA overexpression, excluded mRNAs that didn't show stronger repression following the 8mer>7mer>6mer repression pattern. These selections will already remove many MTS or miRNA targeted mRNAs that are negatively affected by RBPs – hence weaker response to miRNA overexpression. Furthermore, many miRNA target sites can exist in a single mRNA. They didn't explain how they classify these mRNAs into each category when multiple MTS are present in a single mRNA. Altogether, both their RBS and MTS are incomplete and not well-defined. It makes any subsequent analysis very questionable and convoluted.

Second, they have not satisfactorily explained how RBPs, with a much lower binding affinity to RNAs in general than the RISC (~1,000-fold), can open the RNA whereas the RISC couldn't do so. It's conceivable if the RBP has the helicase activity but they are only a small portion of RBPs. Nor did they test this idea directly. Although they provided some interesting binding data in Fig. 5c-d, the specificity of these assay can be further improved such as mutating the miRNA binding sites or using non-targeting miRNA programmed rhAGO2.

Third, the most functionally relevant systems are their PCBP2 and IGFBP1 KO cells. By specifically KO these RBPs, they have established a very nice system to examine how the loss of each RBP changes the secondary structure of mRNAs and impairs miRNA targeting. However, only very low resolution

AGO2-IP followed by mRNA quantification assays was done (Fig. 5f). They should perform DMS-seq and AGO2 CLIP-seq (HITS-CLIP, PAR-CLIP or CLEAR-CLIP, to name a few options) to unequivocally establish the correlation between these two RBPs to miRNA targeting efficiency.

Reviewer #2 (Remarks to the Author):

In the manuscript "Widespread regulatory impact of RNA-binding proteins on microRNA targeting", the groups of Shin and Baek explore whether miRNA-mediated gene silencing is affected on a global scale by RNA binding proteins occupying sites on the same target mRNA. They mine the publicly available collection of eCLIP data for 117 different RBPs and relate them to predicted miRNA target sites (MTS). From the observation that "functional" miRNA target sites MTS are located preferentially at hotspots of RBP binding sites (RBS) they come to the conclusion that globally, RBPs act as enhancers of miRNA targeting, either by modulating RNA structure or creating an otherwise permissive environment for miRNA targeting. Setting aside some potential technical problems, this statement clearly mistakes correlation for causation. The fact that there is much eCLIP signal around miRNA target sites does not mean at all that these RBPs influence miRNA targeting efficiency. They only test this proposition in one Figure of a 7-figure manuscript using a handful of luciferase reporter constructs, which appear to largely confirm their hypothesis.

Evaluation of this manuscript with its hundreds of different datasets is quite difficult, as at times the presentation unfortunately lacks clarity (to some degree understandable considering the complexity of the underlying datasets). That being said, the authors did a great job generating beautiful figures to complement the text.

This manuscript would be a good candidate for Nature Communications, if the authors were able to substantially strengthen their conclusion that RBPs globally aid miRNA targeting by addition of more genome-wide data focusing on RBP loss-of-function coupled with miRNA transfections (with the understanding that this is not possible to do comprehensively for all RBP and miRNA combinations, but rather in a targeted manner). While the collection of data and careful analysis presented is by itself very valuable and may represent a wonderful resource, particularly for refinement of miRNA target site prediction algorithms, adding more high-throughput data and better controls to the already presented experiments to bolster their hypothesis is critical, considering the provocative claim.

Additional comments:

- RBS occupancy: The authors decide to ignore occupancy of RBS binding sites, even though it is clear that many of the examined RBPs do not accumulate in the cytoplasm in copy numbers that are sufficient to result in RNA regulation. It might even be conceivable that some of the "hotspots" of RBS that they observe may be some form of universal background or alternatively that the presence of the MTS itself creates a permissive environment for RBP binding (basically the inverse of the authors' hypothesis).

- Fig. 1a and b: The authors observe that efficiency of miRNA mediated targeting is reduced if the MTS is distal to RBSs (and they observe that this effect is greater if they move away from the cell lines in which the eCLIP data were generated). This raises the question whether the authors took into account that 3' UTRs may vary between different cell lines - e.g. if a 3'UTR is shorter in HeLa than in HepG2, then the RBS that were all determined in HepG2 might not be available.

- Fig. 1b: It would be good to perhaps clarify how the observations were counted: was only the closest RBS to the MTS considered, which I assume? In that case, it is difficult to understand the numbers -

the Friedman/Bartel paper from 2008 finds 500 or so conserved sites per miRNA - and the authors transfect 28 miRNAs, which would result in a (very, very generous) estimation of 25000 or so functional sites the authors could observe. How can there be ~1 M observations for HepG2 alone? Or was an MTS counted multiple times, if there was an RBP on the same 3'UTR? In that case, it is absolutely crucial to understand whether the decay of MTS efficiency with distance of RBS to MTS could be an analytical artifact of 3'UTR length; any MTS that is e.g. located in the middle of a 1k long UTR cannot have RBS that are more distant than 500 nt. Finally, UTR heterogeneity could also confound this analysis, e.g. if only a minority of UTRs are long - they would still have the RBS detected in eCLIP, but the MTS would act on the shorter versions (which in my mind makes considering RBS occupancy for these analyses important).

- In the description for Fig. 1 and 2 the authors make multiple statements along the lines of the one on line 116: "Taken together, these results demonstrate that most RBPs, if not all, enhance MT when binding close to the MTS, while no RBP detectably suppresses MT on a global scale.". At this point in the manuscript this is absolutely not true and mistakes correlation for causation. At this point they can only say that functional MTS are close to regions of binding of other RBPs, as measured by eCLIP. Until they test this hypothesis experimentally (in Fig. 6) it would be advisable to phrase these statements more carefully and possibly phrase them more as a hypothesis.

- Regarding reliability fo the eCLIP: it would be good if they had evidence that the eCLIP sites they are considering are not something of a technical background, a problem that plagues all CLIP techniques. E.g. they could take into account high-affinity binding motifs determined for these RBPs by in vitro methods (e.g. Dominguez, Mol Cell 2018), as for cytoplasmic RBPs, conserved high-affinity binding sites would be anticipated to be functional. Finally, it would be good to discuss how the authors think about copy numbers of RBPs in the cytoplasm and occupancy - it is clear that MTS are more functional if they represent high-affinity miRNP targets (e.g. McGeary, Science, 2019), which would also mean that for an RBP to support miRNA-mediated gene silencing, it needs to reach substantial occupancy on the targeted mRNA, which is extremely difficult to envision for an RBP with low cytoplasmic localization, even if it were relatively highly abundant, which many are not.

- In Fig. 5e and f the authors test whether miRNA targets are bound less to miRNPs after deletion of IGF2BP1 or PCBP2. They however never test for the influence of these RBPs on target mRNA levels. IGF2BP1 at least is considered to stabilize target mRNAs, so it could be conceivable that the lower recovery of miRNP bound RNA could be due to lower mRNA levels. Controls are crucial to allow the authors to make their statement.

- Fig. 6 is crucial as support for the manuscript's message. The luciferase assays combining RBS and MTS mutations in 6a are a good start. There are a few technical issues: a) the authors never show that the RBPs they assume bind close to the MTS in fact do so. b) In Fig. 6a and b there could be normalization issues. It appears that RBS wt sample is normalized to RBS mut and MTS wt to MTS mut (putting both mut samples as 1) - it would be good if all four bars were normalized to let's say WT - in order to monitor the absolute effect of the mutations on luciferase expression. c) in order to rule out translational effects of the proteins, it would be good to measure luciferase mRNA expression.

- Fig. 6c is the only attempt to validate their hypothesis on a transcriptome-wide scale. They separate RNA expression levels from RNAs with MTS proximal and distal to RBS and they find that IGF2BP1 KO results in preferential derepression of MTS with proximal RBS, which is interpreted as IGF2BP1 KO resulting in reduced miRNA target activity. If my interpretation of their description is correct, then the authors ignore here that IGF2BP1 itself has a strong effect on RNA abundance. They need to control for the possibility that they are in fact observing the effect of the RBP on target mRNA abundance.

- In order to bolster the statement that RBPs globally support miRNA targeting, the authors should take advantage of the RBP KO cells (possibly a few more than just IGF2BP1 and PCBP2, but even that would be a start) and transfect different miRNA duplexes into those cells (and control cells) to show

that in the KO cells miRNA targeting is indeed less efficient in a carefully controlled manner.

- miRNA transfection experiments: from the Materials and Methods section it appears that 28 miRNA-duplexes were selected for transfection into cell lines (they mention four different cell lines) - they should add in the supplement the evidence for which datasets exhibit efficient miRNA targeting. I notice from Table S1 that they use different concentrations of miRNA duplexes - do they observe variation in miRNA target repression efficiency? What is the rationale for using different concentrations of miRNA duplexes? They could for example present a collection of cumulative distribution plots, analogous to Grimson et al., Mol Cell 2007 in the Supplement.

- The Materials and Methods section is massive and comprehensive and I applaud the authors for their transparency. That being said, as a reviewer (and reader) it is difficult to find the chapter that is relevant to evaluate the individual figures. As an example, neither in the text describing Fig. 1, nor in the Methods section itself, it is possible to get the information as to how the MTS were determined. I assume they come from TargetScan, but at this point it is not sure. Furthermore, if it is TargetScan, did they use conserved sites, or all sites, etc. For a Methods section this massive, it might be good to have an additional shortened version that relays the minimal necessary information for the reader to understand each figure. And from there refer to the extended Methods section. An alternative would be to present the analyses for each figure separately.

Reviewer #3 (Remarks to the Author):

Factors that affect miRNA targeting is of fundamental importance. RNA binding proteins and RNA structure have been long suspected playing critical roles in regulating miRNA targeting efficacy. While it is commonly accepted that more open RNA structure facilitates miRNA targeting, proteins are supposed to play both positive and negative regulatory roles. The study by Kim et. al. addressed this question and came to an interesting, simple conclusion that RNA binding proteins also normally promote miRNA binding efficacy.

The conclusion is attractive and probably the study does reveal general trends in the correlation of RBP binding and miRNA targeting. However, the data should be more carefully analyzed, in particular, to be scrutinized in the context of RNA structure. My major concern is that, the observations in this study could also be explained by RNA structure environment, which seems making more sense.

The following are my detailed comments/concerns:

1, figure 1, the study started with all RBPs that have been CLIPped, and it was shown that RBPs as a whole do promote miRNA targeting. But, one of my big concerns is that, most of the proteins analyzed here are single-stranded binding proteins. The occurrence of their binding in proximity is actually an indicative of more single-stranded (ssRNA) structure context, which may be the reason of enhanced miRNA targeting, but not because RBP binding promoted miRNA targeting. It was nice in figure 4, that dsRNA-binding proteins also promote miRNA targeting, but what are these dsRNA-binding proteins? What complicates the analysis is that, even canonical dsRNA-binding proteins also bind to ssRNA.

2, line 96. It is not entirely clear how many of the binding sites are preserved (or stable). Usually RBP binding is quite variable in different conditions, e.g., different cell lines. Is the analysis here done with only the preserved binding sites?

3, figure 2a-c only show the q-value, but not the correlation coefficient. If it is too small, the conclusion is questionable here if the q-value is significant. What test was used to calculate the significance?

4, figure 2d, similar to the above point 1, it could be also the results that most proteins are ssRNA-binding proteins.

5, figure 3b, indeed, this figure seems to confirm that RBPs as a whole bind to more single-stranded RNAs. So as mentioned in point 1, the observations in this study could be explained by RNA structure. Another technique issue for this figure and some other panels in figure 3, why use DMS reads? In these RNA structure probing methods, reads cannot always represent the level of RNA structure. It also depends on background structural signals. So normally the DMS-score that more closely represent RNA structures is used.

6, importantly, suggest to also use DMS-seq in vitro data to repeat analysis in Figure 3. The authors claim that "the elevated level of DMS-seq can be attributed to higher RBP-binding activity". I am curious about the DMS-seq structure signal in deproteinized in vitro samples, because in those samples, there is no protein binding.

7, figure 3d, it is argued that RBP binding opens RNA structure. But this is not fully supported. Only correlations were observed. It could also be that RBP tend to bind to more open RNA structures. In another word, more open RNA structure could be the reason, not the consequence. In this way, RNA structure, but not RBP binding, is the true deterministic of miRNA targeting. This constitutes the major concern of this study – how to dissect RNA structure and RBP binding as different contributing factors?

8, figure 5, how come the RNA structural models in panel a? Structure probing should be performed to prove the models, and also for the mutant sequence to see whether the mutations change RNA structures. Also, it is very intriguing that in the model, the miRNA target site are in duplex regions, which will make them bad candidates for miRNA binding. Yet the mutation of RBS further decrease the targeting efficacy.

9, line 197, it says that PCBP2 has not been "reported to interact with AGO". This is nnot "reported not to interact with AGO". Interaction assays are required.

10, line 203, "RBP binding improves target site accessibility of MTSs to AGO both in vitro and in vivo.". RNA structure probing data are need to support this claim.

11, figure 6c, and Extended figure 7a,b, what about results comparing targets and non-targets?

Response to reviewers' comments

We would like to thank the reviewers for their careful and thorough evaluation of this manuscript and for the insightful comments and constructive suggestions, which helped us to improve the quality of the manuscript. Please see below, in blue, our detailed response to reviewers' comments.

Reviewer #1

In this manuscript, Kim *et al.*, examined the impact of RBPs on the efficiency of miRNA targeting. They leveraged a large collection of eCLIP datasets from ENCODE to identify binding sites of 117 RBPs, and correlated their binding (distance and abundance) with the efficiency of miRNA mediated gene repression. They generally observed RBP binding promoting miRNA repression. They provided individual examples for the enhancement of RBP binding to the binding of the RISC *in vitro*. They also knocked out two RBPs, PCBP2 and IGFBP1, and showed Ago2 associated mRNAs were reduced in both cases. Overall, this is a potentially interesting study, leveraging the power of large-scale genomic datasets and bioinformatic analysis to study an important question in RNA biology and gene expression regulation. However, I have several major concerns that should be carefully addressed by the authors.

(R1-1) First, the main conclusion, RBPs globally enhance miRNA targeting, is largely based on extrapolation. Specifically, they used 117 RBP eCLIP datasets to generate the global RBP-binding sites (RBS) information, which is critical to calculate the distance between miRNA binding sites and RBS. However, 117 RBPs are less than 10% of 1,500 RBPs, and it was unknown how many of these 1,500 RBPs are expressed in HepG2 and HeLa cells, the cell source of their eCLIP data. Even with a conservative estimation, 30% of the total RBPs should be expressed at a reasonable level in these cells, which suggest ~450 RBPs can bind to mRNAs, far more than 117 RBPs. Without such knowledge, it is impossible to accurately determine the true distance between any RBS to MTS (miRNA target sites) because any unbound mRNA sequences could be occupied by undetermined RBPs.

In addition, they have not further narrowed down the RBS by searching for any specific binding motif for any RBPs. Without the binding motif information, the definition of RBS is not strong. Both of these two issues are problematic for the relatively long $d_{\text{MTS-RBS}}$ cases such as 500~1k or 1k~10k categories. They could be, in fact, much shorter distance to nearby MTS. Yet, they still observed weaker effect on miRNA repression of these apparently long $d_{\text{MTS-RBS}}$ sites.

Another critical component of this analysis is miRNA target sites (MTS). They have not clearly stated how they identify MTS. Based on the Method section (pg. 54), they selected 28 conserved but not very highly expressed miRNA for overexpression. Then they just used 8mer, 7mer-m8, 7mer-A1 and 6mer, presumably the different matches to the seed sequences of these miRNAs, to define MTS (pg. 54, **line 871**), which is clearly not adequate given the high false positive rates of identifying MTS by the seed match alone. Next, they heavily selected a portion of mRNAs for their analysis – they discarded upregulated genes upon miRNA overexpression, excluded mRNAs that didn't show stronger repression following the 8mer>7mer>6mer repression pattern. These selections will already remove many MTS or miRNA targeted mRNAs that are negatively affected by RBPs – hence weaker response to miRNA overexpression. Furthermore, many miRNA target sites can exist in a single mRNA. They didn't explain how they classify these mRNAs into each category when multiple MTS are present in a single mRNA.

Altogether, both their RBS and MTS are incomplete and not well-defined. It makes any subsequent analysis very questionable and convoluted.

We agree that the 117 RBPs investigated in our study are a relatively minor fraction of total human RBPs as it is reported that about 1,500 RBPs are present in human cells. However, we would like to emphasize that 117 RBPs are the currently available largest dataset of human RBPs and the scale of our analysis is dozens-fold larger than any other previous studies. Furthermore, to alleviate the reviewer's concern, we have additionally analyzed the recently added 55 eCLIP-seq experiments (44 RBPs) that have not been investigated in the current version of our manuscript and therefore can serve as a perfectly independent validation dataset to examine whether our observations can be extrapolated to other RBPs. Accordingly, we have discovered that our conclusions are nicely reproduced: the association between $d_{\text{MTS-RBS}}$ and MT efficacy was consistently observed in this newly added dataset (**Fig. R1-1b**) and the expanded dataset of 150 RBPs (**Fig. R1-1c**) showed clearer association compared to our previous result from 117 RBPs (**Fig. R1-1a**). The association analysis with the number of RBPs nearby the MTSs also exhibited consistent result (**Fig. R1-1**). These results highlight that the observed association between RBP binding and enhanced MT efficacy can be extrapolated further to the comprehensive set of RBPs.

Regarding the binding motifs of RBPs, we have utilized *in vivo* RBP-binding motif database mCross (Feng *et al.*, 2019, Mol. Cell) in our analysis, in order to investigate the association between the number of RBPs bound close to the MTS and MT efficacy with more reliable RBSs supported by binding motifs (**Supplementary Fig. 2e**). In response to the reviewer, we performed the association analysis of **Fig. 1d** with a subset of RBSs that have specific binding motifs and observed consistent association between $d_{\text{MTS-RBS}}$ and MT efficacy (**Fig. R1-1d**), also supporting the robustness of our findings.

As the reviewer mentioned, collection of MTSs by seed matching alone can include a number of false positives. However, when only the MTSs with 8mer site type were examined, which has a significantly reduced false positive rate, the association between RBP binding and enhanced MT efficacy was consistently observed (**Fig. 4a**). To more directly address the reviewer's concern, we iterated the association analysis on a subgroup of MTSs that are strongly enriched for true positives (context score < -0.1) (**Fig. R1-1e**). Likewise, we found that RBP binding was associated with enhanced MT efficacy for a subgroup of MTSs with good context (**Fig. R1-1f**). From these consistent results, although some false positives would be included among the MTSs used in the investigation, we believe that the association we observed is robust and valid. In response to the reviewer, we have improved our manuscript, supplementing the result of association analysis with MTSs of good context at **Supplementary Fig. 5b**.

Regarding the reviewer's concern on mRNA selection, we would like to clarify that we did not discard individual mRNAs upregulated upon miRNA overexpression, but excluded whole transcriptome data when the target mRNAs are upregulated compared to the non-targets, or the target mRNAs do not follow the known hierarchy of average MT efficacies between site types. Since these contrary patterns indicate that the transfected miRNAs did not induce the target repression to a detectable degree and thus the experiment was likely to be failed, it is justified to exclude such low-quality data. For the dataset passed the quality check, we included all of target mRNAs that are downregulated, upregulated, or unchanged upon miRNA overexpression. Therefore, the transcriptome datasets we used are not biased to enrich downregulated mRNAs only and thus the RBPs negatively affecting MT can be detected fairly, although such MT repressor RBPs were not identified in our analyses. To reduce confusion on processing and quality control of our transcriptome data, we have revised the **Supplementary Methods** section.

We also would like to clarify that only the targets with a single MTS were used through our analyses in order to clearly observe the impact of RBPs on a single MTS's efficacy. Thanks to the reviewer, we have improved our manuscript to reduce potential confusion.

Figure R1-1. Association analysis between RBP binding and miRNA targeting efficacy for various subset of MTs or RBSs.

a-c, Venn-diagrams of RBPs available in eCLIP-seq experiments were represented at the top. 168 experiments with 117 RBPs were used in the submitted version of manuscript (**a**), Additionally released 55 eCLIP-seq experiments (**b**), Combined datasets ($n=223$) (**c**). In the middle, association analysis results using three eCLIP-seq datasets mentioned above. At the bottom, association analysis between the number of RBPs bound close to the MTS and MT efficacy for three datasets of RBPs. The mean mRNA fold changes of 3'UTRs obtained from mRNA-seq datasets that monitored the whole-transcriptome response to ectopically introduced miRNAs in HepG2 cell line were shown. Each subgroup was carefully chosen to have statistically indistinguishable features, such as the four known confounding features and the RBP-binding intensity of the nearest RBS among four subgroups. The mean values of confounding features for each group were also displayed (***) $P < 0.001$, n.s.: not significant).

d, Association analysis with a subset of RBSs that have specific binding motifs. 3'UTRs with a single 7, 8mer MTS and RBSs that have specific binding motifs curated from mCross database were selected for the subsequent analysis. Otherwise as in **a**.

e, Distribution of the context scores for all single 7, 8mer MTSs used in the study. A subset of MTSs with good context scores (<-0.1) were selected for the subsequent analysis (blue).

f, Association analysis with a subset of MTSs that have a strong context score less than -0.1 . Otherwise as in **d**.

Supplementary Figure 2e. Association analysis between the number of RBPs bound close to the MTS and MT efficacy with respect to the motif specificity. For each of human cell lines (HepG2, HeLa, and other human cancer cell lines), RBP-binding sites were divided into motif-specific (top) and motif-unspecific RBP-binding sites (bottom) depending on whether or not they contain sequence motifs of each RBP, respectively. Each group was carefully selected to have statistically indistinguishable confounding features among subgroups. mRNA fold changes and values of confounding features were compared among subgroups. mRNA fold changes and values of confounding features were compared between groups (Wilcoxon's rank-sum test), and the error bars represent 95% confidence intervals of the values.

(R1-2) Second, they have not satisfactorily explained how RBPs, with a much lower binding affinity to RNAs in general than the RISC (~1,000-fold), can open the RNA whereas the RISC couldn't do so. It's conceivable if the RBP has the helicase activity but they are only a small portion of RBPs. Nor did they test this idea directly. Although they provided some interesting binding data in **Fig. 5c-d**, the specificity of these assay can be further improved such as mutating the miRNA binding sites or using non-targeting miRNA programmed rhAGO2.

A large number of previous studies have provided multiple lines of evidence supporting the ability of RBPs to mediate the unfolding of RNA secondary structures. According to a study of transcriptome-wide RNA structural probing, mRNAs were observed to be less structured state *in vivo* compared to *in vitro* (Rouskin *et al.*, 2014, Nature). Other studies also have shown that RBPs can induce structural change upon RNA by accumulative binding and unwinding of the secondary structures (Dassi, 2017, Frontiers in molecular biosciences; Flores & Ataide 2018, Frontiers in molecular biosciences). More specifically, unfolding RNA structure and increasing accessibility of miR-221/222 site by PUM1 were experimentally proven through bridge PCR experiment (Kedde *et al.*, 2010, Nature Cell Biology).

RBPs with low binding affinity can open RNA structure via both active and passive mechanisms. Active mechanism corresponds to the function of ATP-dependent helicases, which have been pointed out by the reviewer to take a small portion of RBPs. In addition to the helicases, a number of RBPs have been revealed to function as RNA destabilizing chaperone, or to capture the transient, less stable state of RNAs (Lorsch, 2002, Cell; Rajkowitsch *et al.*, 2007, RNA biology). Once RNA is destabilized by either active or passive mechanism, the state presumably is maintained by binding of additional RBPs.

Consistent with the previous studies, we have observed the increased opening of RNA secondary structure (**Fig. 3b**) and the improvement of MT efficacy (**Fig. 2d**) as the number of bound RBPs increases. To examine if RBPs with helicase activities are critical in the observed phenomenon where RBP binding in close proximity of MTS enhances MT, we have compared the obtained results from association analyses between $d_{\text{MTS-RBS}}$ and MT efficacy for 3'UTRs that are categorized to be bound by helicase or non-helicase RBPs (**Fig. 4d**). The results showed that regardless of the helicase activity, RBP binding enhances MT and this association gets stronger as the distance between RBS and MTS decreases. In response to the reviewer, we have improved the discussion on the opening of RNA structure by RBPs in our manuscript.

To improve the specificity of the EMSA as suggested by the reviewer, we have added another experiment where rhAGO2 is loaded with a non-targeting miRNA (**Fig. R1-2**). Whereas the bound fraction increased when rhAGO2 is loaded with a targeting miRNA, the rhAGO2 loaded with non-targeting miRNA did not affect the level of bound fraction. This result supports our claim that the MT efficiency is improved by RBP binding to the vicinity of MTS and this improvement is specifically observed when rhAGO2 is loaded with a targeting miRNA. Thanks to the reviewer, we have improved our result by including a new condition as a negative control and have added this result (**Supplementary Fig. 6c**).

Figure R1-2. Gel mobility-shift assays for RBS^{WT} and RBS^{MUT} of 3'UTRs with RBPs and rhAGO2 loaded with a targeting or a non-targeting miRNA. (Supplementary Fig. 6c (R1-2b) of updated manuscript)

Gel mobility-shift assays were performed in 4 different conditions: (1) with 3'UTR only, (2) with 3'UTR and rhAGO2, (3) with 3'UTR and RBP, and (4) with 3'UTR, RBP, and rhAGO2. The results with rhAGO2 loaded with a targeting miRNA (**a**) and with a non-targeting miRNA, miR-1 (**b**) are shown. Free RNA bands and the RNA:RBP complex bands are shown as black triangles and blue rectangles, respectively. The band intensities were quantitated and the fraction of the bound to the total RNA:RBP complexes were calculated (n=3). The error bars represent 95% confidence interval of the bound fraction.

(R1-3) Third, the most functionally relevant systems are their PCBP2 and IGF2BP1 KO cells. By specifically KO these RBPs, they have established a very nice system to examine how the loss of each RBP changes the secondary structure of mRNAs and impairs miRNA targeting. However, only very low resolution AGO2-IP followed by mRNA quantification assays was done (**Fig. 5f**). They should perform DMS-seq and AGO2 CLIP-seq (HITS-CLIP, PAR-CLIP or CLEAR-CLIP, to name a few options) to unequivocally establish the correlation between these two RBPs to miRNA targeting efficiency.

We agree with the reviewer that AGO2-CLIP datasets can provide additional evidence to support our proposed mechanism. Therefore, we analyzed AGO2 PAR-CLIP-seq data profiled in a miRNA-transfected condition (Hafner *et al.*, 2010, Cell), and were able to detect increased occupancy of AGO2 for MTSs with strong RBP-binding signal, compared to the MTSs with weak RBP-binding signal (**Fig. 3e**). This result, in conjunction with other results in our manuscript, strongly demonstrates that RBP binding opens up the secondary structure of MTSs (**Fig. 3b-d**), facilitates AGO binding (**Fig. 3e**), and eventually induces effective repression of the target mRNAs (**Figs. 1 and 2**).

Additionally, to address the reviewer's concern, we performed DMS-seq in HEK293T WT and *IGF2BP1* KO cell lines to evaluate our claim that RBPs open up the RNA secondary structures. In WT cells, the stringent RBP binding sites (RBSs) of IGF2BP1 had greater DMS reactivity than that of lenient RBSs which indicates that the structure of these RBSs opens up upon binding of IGF2BP1 (**Fig. R1-3**). This result is consistent with our analysis with publicly available DMS-seq dataset (**Fig. 3b**). Conversely, in IGF2BP1 KO cells this pattern was reversed and DMS reactivity decreased as the RBS signal stringency increased (**Fig. R1-3**). The analysis indicates that when IGF2BP1 is absent, the secondary structure of the RBSs is in more closed state and this structural change is highly specific to the IGF2BP1 binding sites because the DMS reactivity decreased as the RBS signal stringency increased.

Together, AGO2 PAR-CLIP-seq data and DMS-seq data demonstrate that RBPs open up the secondary structure of mRNAs and enhance miRNA targeting. Thanks to the reviewer, we were able to improve our manuscript and have added this result (**Fig. 5e**).

Figure R1-3. Comparison of DMS reactivities in WT and RBP KO conditions. (**Fig. 5e** of the updated manuscript)

DMS reactivities on A and C nucleotides of 3'UTRs were measured by comparing DMS(-) and DMS(+) samples. Corresponding nucleotides were divided into three groups by the RBP binding signal of IGF2BP1 eCLIP-seq dataset (No RBS, Lenient, and Stringent). DMS reactivities between WT and RBP KO were compared (Wilcoxon's rank-sum test).

Reviewer #2

(R2-1) In the manuscript "Widespread regulatory impact of RNA-binding proteins on microRNA targeting", the groups of Shin and Baek explore whether miRNA-mediated gene silencing is affected on a global scale by RNA binding proteins occupying sites on the same target mRNA. They mine the publicly available collection of eCLIP data for 117 different RBPs and relate them to predicted miRNA target sites (MTS). From the observation that "functional" miRNA target sites (MTS) are located preferentially at hotspots of RBP binding sites (RBS) they come to the conclusion that globally, RBPs act as enhancers of miRNA targeting, either by modulating RNA structure or creating an otherwise permissive environment for miRNA targeting. Setting aside some potential technical problems, this statement clearly mistakes correlation for causation. The fact that there is much eCLIP signal around miRNA target sites does not mean at all that these RBPs influence miRNA targeting efficiency. They only test this proposition in one Figure of a 7-figure manuscript using a handful of luciferase reporter constructs, which appear to largely confirm their hypothesis.

Evaluation of this manuscript with its hundreds of different datasets is quite difficult, as at times the presentation unfortunately lacks clarity (to some degree understandable considering the complexity of the underlying datasets). That being said, the authors did a great job generating beautiful figures to complement the text.

This manuscript would be a good candidate for Nature Communications, if the authors were able to substantially strengthen their conclusion that RBPs globally aid miRNA targeting by addition of more genome-wide data focusing on RBP loss-of-function coupled with miRNA transfections (with the understanding that this is not possible to do comprehensively for all RBP and miRNA combinations, but rather in a targeted manner). While the collection of data and careful analysis presented is by itself very valuable and may represent a wonderful resource, particularly for refinement of miRNA target site prediction algorithms, adding more high-throughput data and better controls to the already presented experiments to bolster their hypothesis is critical, considering the provocative claim.

We thank the reviewer for evaluating our manuscript as a good candidate for publication in Nature Communications. To examine our hypothesis thoroughly, we have performed various analyses and experiments which provide clear causative evidence for each step of our proposed mechanism; binding of RBPs opens the RNA secondary structure (**Figs. 3b-d and 5e**) and increases the AGO occupancy at the MTS (**Fig. 5d, f**), resulting in the enhanced MT efficacy (**Fig. 6**). Meanwhile, we agree with the reviewer's comment (see also **R2-5**) that claiming the causative relationship between RBP binding and MT enhancement solely based on the results of **Figs. 1 and 2** is an overstatement. From **Figs. 1 and 2**, we can only state that most RBPs are associated with enhanced MT. Thanks to the reviewer, we have revised the related statements in our manuscript.

Regarding the reviewer's suggestion on genome-wide analysis focusing on RBP loss-of-function coupled with miRNA transfection, the analysis has been already included in our current manuscript (**Fig. 6c**). Explanation for the analysis focusing on the impact of RBP deletion on the MT efficacy in transcriptome-scale, and discussion on its implication were added in response to the point **R2-10** below. This miscommunication might have been caused partially by the lack of clarity of our manuscript, also pointed out by the reviewer. In response to the reviewer, we have revised our manuscript to improve the overall readability and clarity.

Additional comments:

(R2-2) RBS occupancy: The authors decide to ignore occupancy of RBS binding sites, even though it is clear that many of the examined RBPs do not accumulate in the cytoplasm in copy numbers that are sufficient to result in RNA regulation. It might even be conceivable that some of the "hotspots" of RBS that they observe may be some form of universal background or alternatively that the presence of the MTS itself creates a permissive environment for RBP binding (basically the inverse of the authors' hypothesis).

For the RBPs enriched in the nucleus rather than the cytoplasm, the occupancies of their RBSs in mature mRNAs can be relatively low, as the reviewer pointed out. However, based on the observation that RBPs with low cytoplasmic fraction exhibits strong association of $d_{\text{MTS-RBS}}$ with MT efficacy comparable to RBPs with high cytoplasmic fraction (**Fig. 4f**), even binding of RBPs with low cytoplasmic fraction appears to be sufficient enough to induce the enhancement of MT.

Regarding the possibility of alternative interpretation on enriched RBP binding near the MTSs with high efficacy, we would like to emphasize that eCLIP-seq utilized in our analyses captures RBP binding in a WT condition where AGO would bind to MTSs of endogenously expressed miRNAs. However, we examined the MTSs of the ectopically introduced miRNAs, which were apparently absent in the WT cells where eCLIP-seq was performed and thus unable to affect eCLIP-seq results. Therefore, the enrichment of RBP binding near the effective MTSs for the transfected miRNAs cannot be explained by the AGO-mediated recruitment of RBPs. We have updated our manuscript to address the concerns raised by the reviewer.

(R2-3) Fig. 1a and b: The authors observe that efficiency of miRNA mediated targeting is reduced if the MTS is distal to RBSs (and they observe that this effect is greater if they move away from the cell lines in which the eCLIP data were generated). This raises the question whether the authors took into account that 3' UTRs may vary between different cell lines - e.g. if a 3'UTR is shorter in HeLa than in HepG2, then the RBS that were all determined in HepG2 might not be available.

As described in **Methods**, we used a set of representative mRNA isoforms which are not specific to a particular cell line. Although use of mRNA annotation specific to the corresponding cell line can provide more accurate information on the existence of the target sites, the overall association observed by our analyses is not largely dependent on cell-line specific conditions. Our consistent results using the representative isoforms in various biological contexts support our claim of the general regulatory impact of RBP binding on MT efficacy. As the reviewer pointed out, 3'UTRs vary between cells, and this variation can affect MT efficacy by inclusion or exclusion of MTSs between different 3'UTR isoforms (Nam *et al.*, 2014, Mol. Cell). Likewise, cell type-specific MT regulatory mechanism by variation in RBSs for different 3'UTR isoforms is an interesting hypothesis worth investigating, which is unfortunately beyond the scope of the present study. Following the reviewer's suggestion, we have included discussions on the potential effect of cell type-specific 3'UTR isoform variation on MT regulation by RBP binding in the revised manuscript.

(R2-4) Fig. 1b: It would be good to perhaps clarify how the observations were counted: was only the closest RBS to the MTS considered, which I assume? In that case, it is difficult to understand the numbers - the Friedman/Bartel paper from 2008 finds 500 or so conserved sites per miRNA - and the authors transfect 28 miRNAs, which would result in a (very, very generous) estimation of 25000 or so functional sites the authors could observe. How can there be ~1 M observations for HepG2 alone? Or was an MTS counted multiple times, if there was an RBP on the same 3'UTR? In that case, it is absolutely crucial to understand whether the decay of MTS efficiency with distance of RBS to MTS could be an analytical artifact of 3'UTR length; any MTS that is e.g. located in the middle of a 1k long UTR cannot have RBS that are more distant than 500 nt. Finally, UTR heterogeneity could also confound this analysis, e.g. if only a minority of UTRs are long - they would still have the RBS detected in eCLIP, but the MTS would act on the shorter versions (which in my mind makes considering RBS occupancy for these analyses important).

The reason why we have ~1M observations for one cell line is because we have taken into account miRNA target sites (MTSs) in various RBP binding information as the reviewer postulated. The most ideal way of estimating $d_{\text{MTS-RBS}}$ is to measure it only once for each MTS by combining RBSs from all human RBPs. However, it is impossible at the current stage because an exhaustive list of all RBSs in human has not yet been characterized since the current CLIP technology has the limitation that it can capture the interactions for one RBP at a time. Besides, there has been no experimental method to determine a complete picture of protein-RNA interactions for all RBPs at once. Therefore, one of the most feasible ways is to define the distance from an MTS to the nearest RBS for each RBP as $d_{\text{MTS-RBS}}$, and regard this observation as an independent measurement for the MTS and this is how we made observations. It is also appropriate and common method for analyzing eCLIP-seq datasets, that focuses on a single RBP without considering other RBPs.

As the reviewer pointed out, the association of $d_{\text{MTS-RBS}}$ with MT efficacy could be confounded by 3'UTR length, which is known to determine miRNA targeting efficacy. For that reason, we presented the results by definitively ruling out the potential confounding effects including the 3'UTR length (**Fig. 1d**). In the analysis, we carefully selected a group of 3'UTRs that have different $d_{\text{MTS-RBS}}$ but have statistically indistinguishable confounding factors. As the confounding effects potentially caused by the previously known features were removed by fitting with multiple regression model, the concern from the analytical artifact can be alleviated. Our observation was still consistent even after a rigorous correction. In response to the reviewer, we have clarified the description of our analysis in **Supplementary Discussion** to reduce potential confusion by the readers.

(R2-5) In the description for **Fig. 1 and 2** the authors make multiple statements along the lines of the one on **line 116**: "Taken together, these results demonstrate that most RBPs, if not all, enhance MT when binding close to the MTS, while no RBP detectably suppresses MT on a global scale.". At this point in the manuscript this is absolutely not true and mistakes correlation for causation. At this point they can only say that functional MTS are close to regions of binding of other RBPs, as measured by eCLIP. Until they test this hypothesis experimentally (in **Fig. 6**) it would be advisable to phrase these statements more carefully and possibly phrase them more as a hypothesis.

As addressed in response to **R2-1**, we agree that our statement regarding **Figs. 1 and 2** is an overstatement. Following the reviewer's suggestion, we have revised the statements in our manuscript and removed premature claims.

(R2-6) Regarding reliability of the eCLIP: it would be good if they had evidence that the eCLIP sites they are considering are not something of a technical background, a problem that plagues all CLIP techniques. E.g. they could take into account high-affinity binding motifs determined for these RBPs by *in vitro* methods (e.g. Dominguez, Mol Cell 2018), as for cytoplasmic RBPs, conserved high-affinity binding sites would be anticipated to be functional. Finally, it would be good to discuss how the authors think about copy numbers of RBPs in the cytoplasm and occupancy - it is clear that MTS are more functional if they represent high-affinity miRNP targets (e.g. McGeary, Science, 2019), which would also mean that for an RBP to support miRNA-mediated gene silencing, it needs to reach substantial occupancy on the targeted mRNA, which is extremely difficult to envision for an RBP with low cytoplasmic localization, even if it were relatively highly abundant, which many are not.

To investigate association between the number of RBPs bound close to the MTS and MT efficacy with more reliable RBSs supported by binding motifs, we have utilized *in vivo* RBP-binding motif database mCross (Feng *et al.*, 2019, Mol. Cell) in our analysis (**Supplementary Fig. 2e**). According to our results, enrichment of motif-specific RBSs was clearly associated with enhanced MT, while the similar pattern was observed for motif-unspecific RBSs. In response to the reviewer, we also performed the analysis of **Fig. 1d** with motif-specific RBSs and observed consistent association between $d_{\text{MTS-RBS}}$ and MT efficacy (**Fig. R2-6**), also supporting the robustness of our findings. Taking these results and the observed association for the RBPs with low cytoplasmic fraction in **R2-2 (Fig. 4f)** together, although the occupancy of some RBSs can be relatively low due to the motif-specificity or intracellular localization of the RBPs, it is considered sufficient to mediate enhancement of MT on a global scale. Following the reviewer's suggestion, we have included discussion on the RBP occupancy and its influence on MT enhancement in **Supplementary Discussion**.

Figure R2-6. Association analysis only with RBSs narrowed down to specific binding motifs. (Supplementary Fig. 5b of updated manuscript)

The mean mRNA fold changes of 3'UTRs obtained from mRNA-seq datasets that monitored the whole-transcriptome response to ectopically introduced miRNAs in HepG2 cell line were shown. 3'UTRs with a single 7, 8mer MTS and RBSs narrowed down to specific binding motifs curated from the mCross database were selected for the subsequent analysis. Each subgroup was carefully chosen to have statistically indistinguishable features, such as the four known confounding features and the RBP-binding intensity of the nearest RBS among four subgroups. The mean values of confounding features for each group were also displayed (** $P < 0.001$, n.s.: not significant).

(R2-7) In **Fig. 5e and f** the authors test whether miRNA targets are bound less to miRNPs after deletion of IGF2BP1 or PCBP2. They however never test for the influence of these RBPs on target mRNA levels. IGF2BP1 at least is considered to stabilize target mRNAs, so it could be conceivable that the lower recovery of miRNP bound RNA could be due to lower mRNA levels. Controls are crucial to allow the authors to make their statement.

To calculate the relative RNA levels, the mRNA level of each target before (input) and after (IPed RNA) flag-AGO2 immunoprecipitation was obtained by qPCR, although not shown in **Fig. 5g**. Therefore, the relative RNA levels shown in the figure are the fold changes of the IPed RNA levels relative to the input RNA levels. We agree with the reviewer that RBPs themselves could change the target mRNA levels but since this effect would have been reflected in both input and IPed RNA levels, it would have already canceled out, not affecting the final reported relative RNA levels. Thanks to the reviewer we have revised **Methods** and the related legends to avoid confusion.

$$\Delta Ct_{Input} = Ct_{Input} - Ct_{Input} = 0, \quad \therefore \text{Relative RNA level}_{Input} = 2^{-\Delta Ct_{Input}} = 2^0 = 1$$

$$\Delta Ct_{IPed} = Ct_{IPed} - Ct_{Input}, \quad \therefore \text{Relative RNA level}_{IPed} = 2^{-\Delta Ct_{IPed}}$$

Figure 5f, g. AGO2-immunoprecipitation (IP) followed by RT-qPCR.

f, Experimental procedure for the experiment of AGO2-IP followed by RT-qPCR.

g, AGO2-IP followed by western blot and RT-qPCR in AGO2-overexpressed HEK293T parental and RBP KO (PCBP2 or IGF2BP1) cells. Protein levels of AGO2, PCBP2, IGF2BP1, and GAPDH in the input and IPed samples were visualized by the western blotting (left). Relative RNA levels of each input and IPed samples were quantitated in parental and RBP KO cells and normalized to each input sample (right, Wilcoxon's rank-sum test, * $P < 0.05$, ** $P < 0.01$, *** $P < 0.001$, $n = 9$ for PCBP2 and $n = 6$ for IGF2BP1). The relative RNA level of the IPed samples are shown in the figure. 'Combined' indicates the combined result of the measured target RNA levels. The RNA level of *KATNA1*, which has no RBS of PCBP2 or IGF2BP1, was used as a negative control and U6 snRNA-level served as a technical control of AGO2-IP. The error bars represent 95% confidence intervals.

(R2-8) Fig. 6 is crucial as support for the manuscript's message. The luciferase assays combining RBS and MTS mutations in 6a are a good start. There are a few technical issues:

a) The authors never show that the RBPs they assume bind close to the MTS in fact do so.

The 3'UTRs used in EMSA (**Fig. 5a-d**) have been selected from the targets used in the luciferase assay (**Fig. 6a, b**). They are ~100-nt fragments containing MTS and RBSs close to the MTS. Therefore, our EMSA data show that the RBPs do bind to the RBSs that are in the vicinity of MTSs as predicted. We have improved the description of the 3'UTRs used in EMSA experiments in the legend of **Fig. 5**.

b) In **Fig. 6a and b** there could be normalization issues. It appears that RBS^{WT} sample is normalized to RBS^{MUT} and MTS^{WT} to MTS^{MUT} (putting both MUT samples as 1) - it would be good if all four bars were normalized to let's say WT - in order to monitor the absolute effect of the mutations on luciferase expression.

For our luciferase assay, we normalized WT to MTS^{MUT} and RBS^{MUT} to $MTS^{MUT}RBS^{MUT}$ double-mutant condition to solely observe the effect of the RBP on microRNA targeting. If both WT and RBS^{MUT} were normalized to MTS^{MUT} , we would have observed the effect of MT for the WT construct while observing the combined effect of both MT and RBS for the RBS^{MUT} construct, which causes our evaluation to be difficult.

To avoid this issue, we compared RBS^{MUT} and $MTS^{MUT}RBS^{MUT}$ double-mutant conditions to rule out the expression level change caused by factors other than MT for RBS^{MUT} . Although the reviewer's suggestion to normalize all to one condition to see the absolute effect seems intriguing, the results cannot be fully interpreted because of the presence of the confounding factors.

c) In order to rule out translational effects of the proteins, it would be good to measure luciferase mRNA expression.

Following the reviewer's advice, we have measured the mRNA expression level of luciferase genes (Renilla and Firefly luciferase) to rule out the translational effects of the proteins (**Fig. R2-8**). Accordingly, we have concluded that translational effects of the proteins are negligible and not likely have affected the luciferase assay results.

Figure R2-8. Luciferase mRNA expression level measured by RT-qPCR.

a, Design of constructs used for the luciferase reporter assays and RT-qPCR are shown left. Four different designed constructs for each selected 3'UTRs were used; (i) wild type MTS and RBS ($MTS^{WT}RBS^{WT}$), (ii) mutated MTS ($MTS^{MUT}RBS^{WT}$), (iii) mutated RBS ($MTS^{WT}RBS^{MUT}$), and (iv) mutated MTS and RBS ($MTS^{MUT}RBS^{MUT}$). The Renilla luciferase (Rluc) mRNA level was normalized by Firefly luciferase (Fluc) mRNA level. Then, the MTS^{WT} was normalized to the cognate mutated MTS ($MTS^{MUT}RBS^{WT}$) and $MTS^{WT}RBS^{MUT}$ was normalized to $MTS^{MUT}RBS^{MUT}$ samples to obtain the normalized mRNA level. (One-way ANOVA, Dunn's multiple comparison test, $n=2$ or 3). The error bars represent 95% confidence intervals.

b, Design of constructs used for the luciferase reporter assays performed and the expected changes in the secondary structures of mRNAs are shown (top left): parental, RBP KO, and rescue conditions of the deleted RBP are depicted by blue, orange, and green colors, respectively with striped colors representing

the mutated MTS. Otherwise as in a.

(R2-9) Fig. 6c is the only attempt to validate their hypothesis on a transcriptome-wide scale. They separate RNA expression levels from RNAs with MTS proximal and distal to RBS and they find that IGF2BP1 KO results in preferential de-repression of MTS with proximal RBS, which is interpreted as IGF2BP1 KO resulting in reduced miRNA target activity. If my interpretation of their description is correct, then the authors ignore here that IGF2BP1 itself has a strong effect on RNA abundance. They need to control for the possibility that they are in fact observing the effect of the RBP on target mRNA abundance.

Response to this comment **(R2-9)** was addressed along with **R2-10**. See below.

(R2-10) In order to bolster the statement that RBPs globally support miRNA targeting, the authors should take advantage of the RBP KO cells (possibly a few more than just IGF2BP1 and PCBP2, but even that would be a start) and transfect different miRNA duplexes into those cells (and control cells) to show that in the KO cells miRNA targeting is indeed less efficient in a carefully controlled manner.

Given the reviewer's comment **R2-9** and **R2-10**, we noticed that our explanation of the data and the meaning of the result in Fig. 6c was not clear enough and can be misunderstood by readers. For the analysis of Fig. 6c, we generated mRNA-seq data which measured the mRNA expression levels upon transfection of miRNAs in WT and RBP KO cells, respectively (**Supplementary Table 2**). The x-axis indicates the difference of the $\log_2(\text{mRNA fold change})$ values between RBP KO and WT cells, which represents the change of MT efficacy upon the removal of RBP. Accordingly, the right-shifted distribution means that the target mRNAs are less efficiently repressed in RBP KO cells, and we have observed that the target mRNAs with short $d_{\text{MTS-RBS}}$ (orange) exhibited such right-shifted distribution in response to the removal of RBP while the target mRNAs with long $d_{\text{MTS-RBS}}$ (blue) did not. Therefore, our analysis of Fig. 6c is exactly what the reviewer has suggested in **R2-10**. We have improved the description of our analysis and x-axis label of Fig. 6c and **Supplementary Fig. 7a, b**, in order to reduce potential confusion.

As the reviewer pointed out, the effect by the own function of knocked-out RBPs on mRNA abundance can distort the mRNA fold change values. Theoretically, the effect of RBP function independent to the MT would be canceled out when calculating mRNA fold change by comparing miRNA- and mock-transfected conditions within RBP KO cells. Nevertheless, to make sure that our result is not biased by the own function of the RBP on mRNA abundance, we added the CDF of the mRNAs without any MTS in the 3'UTR ('No-site' group), in response to the reviewer's comment (**Fig. 6c** and **Supplementary Fig. 7a, b**). The distribution of 'No-site' mRNAs (gray) was fairly similar to that of the target mRNAs with long $d_{\text{MTS-RBS}}$ (blue). This result indicates that the effect of the knocked-out RBP's own function on mRNA abundance is properly controlled, and the observed de-repression of the miRNA targets with proximal RBS (orange) would be the effect of the RBP on MT efficacy, not influenced by its own function. Thanks to the reviewer, we have updated our analysis of Fig. 6c and **Supplementary Fig. 7a, b**, and improved our manuscript regarding the transcriptome-wide response of miRNA targets after RBP removal.

Figure R2-10. Transcriptome-wide response of miRNA targets after RBP removal. Updated version of Fig. 6c (for 'Combined') and Supplementary Fig. 7a, b (*IGF2BP1* KO and *PCBP2* KO, respectively). Expression fold changes of each mRNA after miRNA overexpression were measured by mRNA-seq for HEK293T parental cells and *IGF2BP1* or *PCBP2* KO cells. The x-axis indicates the difference of the \log_2 (mRNA fold change) values between RBP KO and parental cells, which represents the change of MT efficacy upon KO of RBP. Target mRNAs which contain a single 7, 8mer MTS in their 3'UTRs were selected if $d_{\text{MTS-RBS}}$ is short ($d_{\text{MTS-RBS}} < 100$, orange) or long ($d_{\text{MTS-RBS}} \geq 100$, blue). After controlling for confounding features of MT (right), distributions of \log_2 (mRNA fold change) by KO of RBP were compared between the subgroups by Kolmogorov-Smirnov test (left). The error bars represent 95% confidence intervals. Distribution of miRNA non-targets ('No-site' group, gray) was also plotted and compared after correcting for confounding features.

(R2-11) miRNA transfection experiments: from the **Materials and Methods** section it appears that 28 miRNA-duplexes were selected for transfection into cell lines (they mention four different cell lines) - they should add in the supplement the evidence for which datasets exhibit efficient miRNA targeting. I notice from **Table S1** that they use different concentrations of miRNA duplexes - do they observe variation in miRNA target repression efficiency? What is the rationale for using different concentrations of miRNA duplexes? They could for example present a collection of cumulative distribution plots, analogous to Grimson *et al.*, Mol Cell 2007 in the Supplement.

To find optimal transfection concentration of the designed miRNA duplexes, we transfected 100nM, 200nM and 300nM of the duplexes, assuming each designed duplex may have different targeting efficiency depending on the transfection concentration. Then, we selected the experiments that passed the QC criteria mentioned in **Supplementary Methods**. To observe miRNA targeting efficacy by the transfected miRNA duplexes at different concentrations, following the reviewer's suggestion, we have plotted cumulative distribution functions of mRNA fold changes with respect to transfection concentrations and site types: a single 8mer, 7mer-m8, 7mer-A1, and 6mer. In terms of the extent of target repression and site type hierarchy, we have observed no noticeable variation with respect to different concentrations (**Fig. R2-11**), and therefore the averaged expression levels across different concentrations was used when comparing with mock samples.

Figure R2-11. miRNA targeting efficacy and different concentrations for miRNA duplexes.

a, Cumulative distribution functions by different concentrations of duplexes (100nM, 200nM, and 300nM). mRNA targets for corresponding miRNA are colored by their site types: blue for a single 8mer, magenta for 7mer-m8, orange for 7mer-A1, and green for 6mer.

b, Scatter plots for comparison of mRNA expression levels from hsa-miR-211-5p transfected samples with various transfection conditions. No noticeable difference among the experiments with various concentrations of miRNA duplexes was observed. Spearman's rho was calculated for representing the correlation of two different samples.

(R2-12) The **Materials and Methods** section is massive and comprehensive and I applaud the authors for their transparency. That being said, as a reviewer (and reader) it is difficult to find the chapter that is relevant to evaluate the individual figures. As an example, neither in the text describing **Fig. 1**, nor in the Methods section itself, it is possible to get the information as to how the MTS were determined. I assume they come from TargetScan, but at this point it is not sure. Furthermore, if it is TargetScan, did they use conserved sites, or all sites, etc. For a Methods section this massive, it might be good to have an additional shortened version that relays the minimal necessary information for the reader to understand each figure. And from there refer to the extended Methods section. An alternative would be to present the analyses for each figure separately.

We would like to thank the reviewer for the valuable comments. We have changed the **Methods** sections to address the reviewer's comment. The sections have been modified to a shortened version containing minimal information for understanding of each figure and the remaining sections have been moved to **Supplementary Methods**. We hope this modification makes it easier to find contents relevant to each figure.

Reviewer #3

Factors that affect miRNA targeting is of fundamental importance. RNA binding proteins and RNA structure have been long suspected playing critical roles in regulating miRNA targeting efficacy. While it is commonly accepted that more open RNA structure facilitates miRNA targeting, proteins are supposed to play both positive and negative regulatory roles. The study by Kim et. al. addressed this question and came to an interesting, simple conclusion that RNA binding proteins also normally promote miRNA binding efficacy.

The conclusion is attractive and probably the study does reveal general trends in the correlation of RBP binding and miRNA targeting. However, the data should be more carefully analyzed, in particular, to be scrutinized in the context of RNA structure. My major concern is that, the observations in this study could also be explained by RNA structure environment, which seems making more sense.

The following are my detailed comments / concerns:

(R3-1) Figure 1, the study started with all RBPs that have been CLIPped, and it was shown that RBPs as a whole do promote miRNA targeting. But, one of my big concerns is that, most of the proteins analyzed here are single-stranded binding proteins. The occurrence of their binding in proximity is actually an indicative of more single-stranded (ssRNA) structure context, which may be the reason of enhanced miRNA targeting, but not because RBP binding promoted miRNA targeting. It was nice in **Figure 4**, that dsRNA-binding proteins also promote miRNA targeting, but what are these dsRNA-binding proteins? What complicates the analysis is that, even canonical dsRNA-binding proteins also bind to ssRNA.

The reviewer raised a concern that the observed enhancement of miRNA targeting (MT) efficacy can be the consequence of the already single-stranded RNA structure context, rather than induced by the opening of structured RNA resulting from the RBP binding. To address this issue, we additionally used *in vitro* DMS-seq data which provides us structural information for RNAs of RBP-unbound state. Following the reviewer's suggestion, we calculated DMS scores from each data and compared the scores between *in vivo* and *in vitro* (See also our response to reviewer's comment **R3-5** for more details on the calculation). The change in DMS score indicates whether RNAs in living cells are more single-stranded or not. As a result, the DMS scores for 3'UTR fragments containing RBSs increased *in vivo* condition compared to *in vitro*, and the level of increment was higher than that of 3'UTR fragments without any RBP binding ('No RBS' group) even after careful control of confounding factors (**Fig. R3-1a**). The enhancement of *in vivo* DMS score was more pronounced when a 3'UTR fragment contains stringent RBSs or a larger number of RBPs bind to the fragment (**Fig. R3-1a, b**). When examining individual RBPs, the enhancement of *in vivo* DMS score was detected and the levels of enhancement were higher than 'No RBS' group for most of RBPs (119 of 120 RBPs, **Fig. R3-1c**). These results provide evidence that the RBP binding can induce the opening of the structured RNAs, and this opening can be mediated by almost all RBPs.

For analysis in **Fig. 4e**, we collected gene ontology annotations for the RBPs in eCLIP-seq dataset and found 7 and 9 RBPs are explicitly annotated as single-stranded RNA (ssRNA) binding and double-stranded RNA (dsRNA) binding, respectively. The annotated dsRNA-binding proteins are DGCR8, DHX30, DROSHA, HNRNPU, ILF3, SUPV3L1, TARDBP, XRCC6, and YBX3. The consistent associations for subsets of RBPs separated by strand specificity indicate that the binding of RBPs enhances MT efficacy regardless of their strand specificity, and can alleviate the reviewer's concern about the potential bias due to the dominance of ssRNA-binding proteins in the eCLIP-seq dataset (**Fig. 4e**).

The mechanism of how the binding of dsRBPs enhances MT efficacy is probably through their dual

capability of binding both ssRNAs and dsRNAs. For instance, it is known that DGCR8, one of the dsRBPs that we analyzed, binds ssRNAs and dsRNAs with similar affinities (Roth *et al.*, 2013, JBC). eCLIP-seq analysis based on UV-induced crosslinking technique mainly capturing the single-stranded interactions (Wheeler *et al.*, 2018, WIRES RNA) revealed that dsRBPs also have >10,000 3'UTR RBSs (**Supplementary Fig. 1a**). Our DMS-seq analysis also indicates that binding of the RBPs generally induces the opening of the secondary structure, regardless of strand specificity (**Fig. R3-1c**). Taken together, these results indicate that RBP binding can open the RNA structure regardless of the own functions and/or strand-specificity of RBPs, and therefore our model can be generalized when understanding the impact of RBPs on MT. In response to the reviewer, we have included an updated version of DMS-seq analysis in our manuscript and improved the description on general mechanism of MT enhancement by binding of various RBPs.

Figure R3-1. Revised DMS-seq analysis by use of DMS score for precise measurement of RNA structures. Updated version of Fig. 3b-d.

a, Δ DMS score representing how much RNA structure open *in vivo* relatively to *in vitro* was used. To make sure the RNA structure of RBP-unbound state, DMS score *in vitro* was utilized instead of minimum free energy (MFE) of *in silico* predicted RNA structure. Two factors (DMS score *in vitro* and mRNA level) were controlled as being indistinguishable across three groups: No RBS, Lenient, and Stringent.

b, Correlation between the number of RBPs and change of DMS reactivity *in vivo* (Δ DMS score). Δ DMS scores of the first and last bins were compared (Wilcoxon's rank-sum test).

c, Change of DMS reactivity *in vivo* (Δ DMS score) for individual RBPs. 'No RBS' group consists of 3'UTR fragments without binding of any RBPs profiled in K562 cell line. For each group of fragments which contain specific RBP binding, Δ DMS scores were compared to that of 'No RBS' group (Wilcoxon's rank-sum test).

(R3-2) Line 96, It is not entirely clear how many of the binding sites are preserved (or stable). Usually RBP binding is quite variable in different conditions, e.g., different cell lines. Is the analysis here done with only the preserved binding sites?

Following the reviewer's suggestion, we have newly identified the preserved binding sites between two cell lines (K562 and HepG2) and performed the same association analysis as in **Fig. 1d**. We have defined the preserved binding sites as the intersection of eCLIP-seq peaks called in the two cell lines (**Fig. R3-2a**). We have calculated the fraction of preserved binding sites for each RBP: ~29% and ~23% for K562 and HepG2 on average, respectively (**Fig. R3-2b**). Consistent with the result using all RBPs, the association analysis with only the preserved RBSs also indicated that the features that we discovered ($d_{\text{MTS-RBS}}$, Num. of RBPs) still showed significant association with the miRNA efficacy (**Fig. R3-2c, d**). Thanks to the reviewer for the suggestion, we have added the results to our supplementary figures (**Supplementary Figs. 1g and 2c**).

Figure R3-2. Association analysis between RBP binding and miRNA targeting efficacy using preserved binding sites. (Supplementary Fig. 1g (R3-2c) and Supplementary Fig. 2c (R3-2d) of updated manuscript)

a, Two examples showing the definition of preserved binding sites used in the analysis. RBP-binding sites (RBSs, eCLIP-seq peaks) from two cell lines and two replicates are represented as thin features. Union region from two replicates for each cell line is denoted as gray box. Preserved sites derived from intersection of two gray boxes are represented as thick features.

b, The fractions of preserved RBSs for overall and individual RBPs in two cell lines. Orange and blue bars represent the fraction of the RBSs in HepG2 and K562 respectively.

c, Association analysis between the $d_{\text{MTS-RBS}}$ and MT efficacy using the preserved binding sites.

d, Association analysis between the number of RBPs bound close to the MTS and MT efficacy using preserved binding sites.

(R3-3) Figure 2a-c only show the q-value, but not the correlation coefficient. If it is too small, the conclusion is questionable here if the q-value is significant. What test was used to calculate the significance?

In response to the reviewer, we have obtained correlation coefficients for individual RBPs by comparing the distance we suggested ($d_{\text{MTS-RBS}}$) with MT efficacy ($\log_2(\text{mRNA fold change})$). Consistent with the result in the manuscript (**Supplementary Fig. 1c**), averaged correlation coefficient for the $d_{\text{MTS-RBS}}$ (0.12) was greater than that of target abundance (TA, 0.077) or seed pairing stability (SPS, 0.078) previously discovered as important determinants for MT (**Fig. R3-3**). The result provides strong evidence that RBP binding is a primary determinant of MT, even compared to the known determinants.

As we have already described in our **Methods**, the statistical tests used in **Fig. 2a-c** were the two-tailed t -tests for the parameter $d_{\text{MTS-RBS}}$ in the model (`OLS.fit()` in Python statsmodels package), followed by false discovery rate correction. The resultant q values indicate how the parameter (independent variable) was significantly associated with the MT efficacy (outcome variable).

Figure R3-3. Correlation coefficients of parameters. Parameters such as target abundance (TA, top), seed pairing stability (SPS, middle) and distance between miRNA target site and RBP binding site ($d_{\text{MTS-RBS}}$, bottom), were independently compared to $\log_2(\text{mRNA fold change})$ values to obtain correlation coefficients (Spearman's rho, ρ). Averaged coefficients were indicated by dashed lines and figures on the

right. $\log_2(\text{mRNA fold change})$ values monitored in HeLa cells were used.

(R3-4) Figure 2d, similar to the above point 1, it could be also the results that most proteins are ssRNA-binding proteins.

As addressed in response to the comment **R3-1**, our results suggest that RBPs bind to RNA and open the RNA secondary structure. Therefore, we suggest the association between enriched binding of RBPs near the MTSs and enhanced MT efficacy shown in **Fig. 2d** is the result that RBP binding enhances MT efficacy by opening the RNA structure and increase AGO accessibility, supported by multiple lines of evidence from the followed analyses and experiments.

(R3-5) Figure 3b, indeed, this figure seems to confirm that RBPs as a whole bind to more single-stranded RNAs. So as mentioned in point 1, the observations in this study could be explained by RNA structure. Another technique issue for this figure and some other panels in **Figure 3**, why use DMS reads? In these RNA structure probing methods, reads cannot always represent the level of RNA structure. It also depends on background structural signals. So normally the DMS-score that more closely represent RNA structures is used.

As the reviewer suggested, we replaced the DMS reads with DMS score for more accurate indication of RNA structural signal. Here is a brief method how we obtained DMS score (**Fig. R3-5**), which is similar to that of the previous study (Berkowitz *et al.*, BMC bioinformatics, 2016). We have also supplemented **Methods** with the detailed description. First, we used normalized DMS read counts to RPM scale for each replicate of DMS-seq data in K562 (Rouskin *et al.*, Nature, 2014). The DMS levels were then calculated by averaging the normalized counts of multiple replicates. Second, we removed the background signals by comparing DMS levels between *in vivo* and denatured control and defined the resulting values as 'DMS score' (**Eq. 1**). DMS score for *in vitro* sample was also calculated in the same way (**Eq. 2**). Third, we introduced new metric ' Δ DMS score' by subtracting DMS score *in vitro* from DMS score *in vivo* for each nucleotide (**Eq. 3**). Δ DMS score for a 3'UTR fragment was calculated as an average score for nucleotides within the fragment.

We also improved the process of controlling confounding factors such as RNA structures of RBP-unbound state by additionally replacing minimum free energy (MFE) with *in vitro* DMS score, which more directly reflects the secondary structure of RNAs in the deproteinized state. Even when we controlled for *in vitro* structure and mRNA expression level among three groups (No RBS, Lenient, and Stringent), we were still able to observe the structural changes caused by the RBP binding. Thanks to the reviewer, we have revised our DMS-seq analysis of **Fig. 3b-d**, strengthening our message for causal relationship of RBP binding and structural changes.

$$DMS\ score_{in\ vivo} = \log_2 \left(\frac{DMS\ level_{in\ vivo}}{DMS\ level_{denatured}} \right) \dots \dots \dots (Equation\ 1)$$

$$DMS\ score_{in\ vitro} = \log_2 \left(\frac{DMS\ level_{in\ vitro}}{DMS\ level_{denatured}} \right) \dots \dots \dots (Equation\ 2)$$

$$\Delta DMS\ score = DMS\ score_{in\ vivo} - DMS\ score_{in\ vitro} \dots \dots \dots (Equation\ 3)$$

Figure R3-5. The calculation of DMS score and its change.

An example of 100-nt 3'UTR fragment for representing the calculation of DMS score. Positions in 1,297-1,396 of the mRNA transcript GAPDH (NM_002046) were plotted. DMS levels *in vitro*, *in vivo* and in denatured conditions were normalized in RPM (top three rows). Then we used it for averaging DMS scores and their relative changes *in vivo* compared to *in vitro* (Δ DMS scores) of the individual 3'UTR fragment (bottom two rows).

(R3-6) Importantly, suggest to also use DMS-seq *in vitro* data to repeat analysis in **Figure 3**. The authors claim that “the elevated level of DMS-seq can be attributed to higher RBP-binding activity”. I am curious about the DMS-seq structure signal in deproteinized *in vitro* samples, because in those samples, there is no protein binding.

In our previous analyses, we have mainly monitored the DMS reads of *in vivo* data and controlled the confounding effect of RNA structure in absence of RBPs by using the minimum free energies of the predicted structures. Inspired by the reviewer's suggestion, we additionally utilized the DMS-seq data of denatured RNAs and *in vitro* condition in our analyses, which correspond to the basal level of DMS reactivity for each nucleotide position and RNA secondary structure of deproteinized environment, respectively. Accordingly, as described in response to the comment **R3-5**, we have modified our analyses by measuring the change in DMS score between *in vivo* and *in vitro* conditions and used *in vitro* DMS score to control the confounding effect. As addressed in response to the comment **R3-1**, significant enhancement of the DMS score was observed for the 3'UTR fragments when RBPs bind, compared to the fragments without binding of any RBP but with similar DMS scores of *in vitro* condition (**Fig. R3-1a**). Taken together with the previous study which already observed the less structured state of mRNAs *in vivo* (Rouskin *et al.*, 2014, Nature), the opening of the RNA structure is considered to be mediated by binding of RBPs in the cells. Thanks to the reviewer, we have improved our manuscript regarding the impact of RBP binding on RNA structure.

(R3-7) Figure 3d, it is argued that RBP binding opens RNA structure. But this is not fully supported. Only correlations were observed. It could also be that RBP tend to bind to more open RNA structures. In another word, more open RNA structure could be the reason, not the consequence. In this way, RNA structure, but not RBP binding, is the true deterministic of miRNA targeting. This constitutes the major concern of this study – how to dissect RNA structure and RBP binding as different contributing factors?

As we addressed in response to the comments **R3-1, 5, and 6**, we demonstrated that RBP binding directly drives opening of RNA structure (**Figs. 3b-d and R3-5**). After controlling *in vitro* RNA structures to be similar between groups of 3'UTR fragments, we clearly observed the opening of RNA structure for RBP-bound 3'UTR fragments *in vivo*. These results confirm the causative relationship between RBP binding and open RNA structure.

To more explicitly address the reviewer's concerns about the causation of RBP binding and RNA structure, we performed DMS-seq in HEK293T WT and *IGF2BP1* KO cell lines to substantiate our claim that RBPs open up the RNA secondary structures. In WT cells, the stringent RBP binding sites (RBSs) of *IGF2BP1* had greater DMS reactivity than that of lenient RBSs which indicates that the structure of these RBSs opens up upon binding of *IGF2BP1* (**Fig. R3-7**). This result is consistent with our analysis with publicly available DMS-seq dataset (**Fig. 3b**). Conversely, in *IGF2BP1* KO cells this pattern was reversed and DMS reactivity decreased as the RBS stringency increased (**Fig. R3-7**). The analysis indicates that when *IGF2BP1* is absent, the secondary structure of the RBSs is in closed state and this structural change is highly specific to the *IGF2BP1* binding sites because the DMS reactivity decreased as the RBS stringency increased. Together, AGO2 PAR-CLIP-seq data and DMS-seq data demonstrate that RBPs open up the secondary structure of mRNAs and enhance miRNA targeting. Thanks to the reviewer, we were able to improve our manuscript and have added this result (**Fig. 5e**).

Furthermore, our result of **Supplementary Fig. 5a** can address the reviewer's concern about dissecting the impact of RNA structure and RBP binding on MT efficacy. For each RBP, multiple linear regression models were fitted by using the changes of secondary structures upon miRNA binding in RBP-absent (ternary interaction model) or RBP-present condition (our proposed model) as a feature (**Fig. 3f**) with the previously known features. The ternary interaction model describes the effect of RNA structure without any influence of RBP binding, while our proposed model describes the effect of RNA structure under the influence of RBP binding. When comparing the two models, our proposed model better explained MT efficacy for multiple RBPs (**Supplementary Fig. 5a**), demonstrating RBP binding is a primary determinant of MT beyond the effect of the structure of RNA itself. Thanks to the reviewer, we revised the manuscript in order to clarify our message from the analysis of **Supplementary Fig. 5a**.

Figure R3-7. Comparison of DMS reactivities in WT and RBP KO conditions. (Fig. 5e of the updated manuscript)

DMS reactivities on A and C nucleotides of 3'UTRs were measured by comparing DMS counts of DMS(-) and DMS(+) samples. Corresponding nucleotides were divided into three groups by the RBP binding signal of IGF2BP1 eCLIP-seq dataset (No RBS, Lenient, and Stringent). DMS reactivities between WT and RBP KO were compared (Wilcoxon's rank-sum test).

(R3-8) Figure 5, how come the RNA structural models in panel **a**? Structure probing should be performed to prove the models, and also for the mutant sequence to see whether the mutations change RNA structures. Also, it is very intriguing that in the model, the miRNA target sites are in duplex regions, which will make them bad candidates for miRNA binding. Yet the mutation of RBS further decrease the targeting efficacy.

The RNA structural models depicted in **Fig. 5a** were generated by *in silico* prediction of RNA secondary structure. For mutant sequences used in our gel-mobility assays (**Fig. 5c, d** and **Supplementary Fig. 6c**), we carefully introduced the mutations while preserving the original secondary structure of the wild-type. Specifically, we randomly shuffled the sequences in specified regions (**Fig. 5a**, orange boxes) to generate the candidates for mutant sequences. Among the shuffled sequences, we selected mutated sequences with the smallest changes of minimum free energy (MFE) values and local AU content compared to those of the structure of RBS^{WT} construct (**Fig. R3-8**).

As noted by the reviewer, the miRNA target sites (MTSs) from the RNA structural models in **Fig. 5a** are inaccessible to AGO2 because of the secondary structure. As mentioned earlier, we mutated the RBSs while preserving the secondary structure (**Fig. R3-8**) to only examine the change in accessibility of miRNA binding sites (MTSs) to AGO2 upon RBP binding. For both constructs, the inaccessibility to AGO2 have been observed in our EMSA result in **Fig. 5d**. These constructs were used to validate our claim that RBP binding alters the secondary structure and enhance accessibility to AGO2 through EMSA (**Fig. 5a-d**) and reporter assays (**Fig. 6a, b**). Thanks to the reviewer, we have supplemented the predicted RNA structures to **Supplementary Fig. 6a**.

Figure R3-8. Predicted secondary structures of RBS^{WT} and RBS^{MUT} constructs.

The secondary structures and MFE values of 3'UTR fragments in SSB (a), UBA1 (b), CRAT (c) were predicted by RNAFold. The constructs including RBS^{WT} and RBS^{MUT} were colored in black and red, respectively.

(R3-9) Line 197, it says that PCBP2 has not been “reported to interact with AGO”. This is not “reported not to interact with AGO”. Interaction assays are required.

As suggested by the reviewer, we conducted interaction assay to identify the relationship between AGO2 and PCBP2. After overexpression of Flag-AGO2, cells were lysed and Flag-AGO2 was immunoprecipitated either after RNase A or mock treatment (**Fig. R3-9**). If the protein of interest is an AGO2-interactor, it would have been co-immunoprecipitated with AGO2 whether RNase A is treated or not. Conversely, if it is not an interactor, it would have been immunoprecipitated less after RNase A treatment. Our western blot data suggests that PCBP2 is not an AGO2 interactor because it is immunoprecipitated less when RNase A is treated to the lysate. This result is in accordance with the results from previous studies that identified PCBP2 or hnRNP E2, as an AGO2 non-interactor (Portnoy *et al.*, 2016, Cell Research, Eiring *et al.*, 2010, Cell, Frohn *et al.*, 2012, Molecular & Cellular Proteomics, Höck *et al.*, 2007, EMBO Rep.).

Also, when we examined the interaction between AGO2 and IGF2BP1, our result indicated that IGF2BP1 is not an AGO2 interactor. This result is consistent with the previous study that showed that IGF2BP1 or ZBP1, do not interact with AGO2 (Höck *et al.*, 2007, EMBO Rep.).

As a positive control, we have included Dicer, which is a well-known component of RISC loading complex and thus is an AGO2 interactor. Additionally, we included PABP1, previously identified AGO2 non-interactor as a negative control (Srivastava *et al.*, 2015, Nature Communications). Thanks to the reviewer, we were able to fortify our claims with a new experimental result and added this result in the manuscript (**Supplementary Fig. 6b**).

Figure R3-9. Co-immunoprecipitation of PCBP2 and IGF2BP1 after RNase A treatment. (Supplementary Fig. 6b of updated manuscript)

Flag-AGO2 overexpressed HEK293T cells were lysed, treated either with RNase A or without, and then immunoprecipitated. The co-immunoprecipitated DICER, PABP1, IGF2BP1, PCBP2, and GAPDH were detected by western blotting. DICER and PABP1, known AGO2 interactor and non-interactor, respectively, were used as controls.

(R3-10) Line 203, “RBP binding improves target site accessibility of MTSs to AGO both in vitro and in vivo.”. RNA structure probing data are need to support this claim.

To support our claim that RBP binding improves target site accessibility of MTSs to AGO *in vitro* and *in vivo*, we have performed EMSA and AGO2-IP qPCR (**Fig. 5**). Through EMSA, we demonstrated that the presence of RBPs can enhance the binding of AGO2 to the MTSs that are inaccessible due to the secondary structure (**Fig. 5d**). Whereas the bound fraction was very low when only 3'UTR and rhAGO2 were present, the bound fraction increased significantly when 3'UTR, rhAGO2, and RBP were present in the reaction. Furthermore, this increase in bound fraction was not as significant when the RBP binding sites (RBSs) of the 3'UTRs have been mutated to reduce the RBP binding. This specificity further supports our claim that the RBP binding improves target site accessibility.

We also have added another experiment where rhAGO2 is loaded with a non-targeting miRNA (**Fig. R3-10a**). Whereas the bound fraction increased when rhAGO2 is loaded with a targeting miRNA, the rhAGO2 loaded with non-targeting miRNA did not affect the level of bound fraction. This result supports our claim that the MT efficiency is improved by RBP binding to the vicinity of MTS and this improvement is specifically observed when rhAGO2 is loaded with a targeting miRNA.

To show RBP binding improves target site accessibility *in vivo*, we have conducted AGO2-IP qPCR in

HEK293T WT and *PCBP2* or *IGF2BP1* KO cell lines (Fig. 5f, g). When the relative level of mRNA with *PCBP2* or *IGF2BP1* binding sites was measured, the relative level of RNA bound by AGO2 increased significantly when *PCBP2* or *IGF2BP1* is present. Therefore, our result indicates that the AGO binds more preferentially to its target mRNA when *PCBP2* or *IGF2BP1* exists in the cell.

Furthermore, we have conducted DMS-seq in HEK293T WT and *IGF2BP1* KO cell lines to examine the structural changes upon RBP binding *in vivo* as suggested by the reviewer (Fig. R3-7). Altogether, our EMSA, AGO2-IP qPCR, and DMS-seq data demonstrate that RBPs open up the secondary structure of mRNAs and enhance miRNA targeting. Thanks to the reviewer, we were able to improve our manuscript by including these additional results (Supplementary Fig. 6c and Fig. 5e).

Figure R3-10. Gel mobility-shift assays for RBS^{WT} and RBS^{MUT} of 3'UTRs with RBPs and rhAGO2 loaded with a non-targeting miRNA. (Supplementary Fig. 6c of updated manuscript)

Gel mobility-shift assays were performed in 4 different conditions: (1) with 3'UTR only, (2) with 3'UTR and rhAGO2, (3) with 3'UTR and RBP, and (4) with 3'UTR, RBP, and rhAGO2. The results with a non-targeting miRNA (miR-1) are shown. Free RNA bands and the RNA:RBP complex bands are shown as black triangles and blue rectangles, respectively.

(R3-11) Figure 6c, and Extended Figure 7a, b, what about results comparing targets and non-targets?

Following the reviewer's suggestion, we revised our analysis of Fig. 6c and Supplementary Fig. 7a, b, by adding CDF for the mRNAs without any MTS in the 3'UTR ('No-site' group) after controlling for confounding features (Fig. R3-11). The distribution of 'No-site' mRNAs (gray) was similar to that of the target mRNAs with long $d_{\text{MTS-RBS}}$ (blue), while the short $d_{\text{MTS-RBS}}$ group (orange) was right-shifted compared to the other two groups. This result confirms that the observed decrease in MT efficacy of the targets with short $d_{\text{MTS-RBS}}$ upon KO of RBP is not affected by the influence of RBP on mRNA abundance. The reviewer's suggestion helped greatly in improving the significance of our research, and we have updated our manuscript regarding the transcriptome-wide response of miRNA targets after RBP removal.

Figure R3-11. Transcriptome-wide response of miRNA targets after RBP removal. Updated version of Fig. 6c (for 'Combined') and Supplementary Fig. 7a, b (*IGF2BP1* KO and *PCBP2* KO, respectively). Expression fold changes of each mRNA after miRNA overexpression were measured by mRNA-seq for HEK293T parental cells and *IGF2BP1* or *PCBP2* KO cells. The x-axis indicates the difference of the $\log_2(\text{mRNA fold change})$ values between RBP KO and parental cells, which represents the change of MT efficacy upon KO of RBP. Target mRNAs which contain a single 7, 8mer MTS in their 3'UTRs were selected if $d_{\text{MTS-RBS}}$ is short ($d_{\text{MTS-RBS}} < 100$, orange) or long ($d_{\text{MTS-RBS}} \geq 100$, blue). After controlling for confounding features of MT (right), distributions of $\log_2(\text{mRNA fold change})$ by KO of RBP were compared between the subgroups by Kolmogorov-Smirnov test (left). The error bars represent 95% confidence intervals. Distribution of miRNA non-targets ('No-site' group, gray) was also plotted and compared after correcting for confounding features.

REVIEWER COMMENTS

Reviewer #1 (Remarks to the Author):

In the revised manuscript, the authors have addressed some of my initial concerns and improved the clarity. However, it is still quite convoluted. It is important to note that the average effect for the strongest effect (overlapped or 1~20nts) was $\sim 2-0.18$, roughly 12% downregulation, whereas for the weakest effect (1~10k) was $2-0.07$, roughly 5% downregulation (Fig. 1b). Although these results may be statistically significant, it is unclear how many of individual cases have measurable effect. I caution that it is inappropriate to extrapolate the conclusion to all of $>1,500$ RBPs, based on these analyses. I also suggest the authors change the very strong claim throughout the manuscript, such as "widespread regulatory impact of $>1,500$ RBPs on MT" in the abstract; "the observed effect of RBPs on MT is undoubtedly a phenomenon occurring in an endogenous environment" pg.5 line 105.

To further improve the clarity, the authors should clearly explain exactly which miRNAs and RBPs are used to generate many panels such as Fig. 1b, 1d, 2d etc, and how the number of observations were calculated. They should also use individual miRNAs (2-5 of them) to generate the panels similar to 1b, and examine whether the general observation still holds true for single miRNA.

Reviewer #2 (Remarks to the Author):

The authors comprehensively engaged with all reviewers' comments and managed to convincingly alleviate most of the concerns.

Their basic observation that increased RBP binding around MTS is correlated with higher efficiency of miRNA silencing appears solid. The explanation that RBP binding prevents RNA secondary structure formation and thus enhanced access to the MTS site is convincing and the likely explanation. Nevertheless, I find that there is no convincing explanation offered as to why dsRBPs or nuclear RBP binding would result in increased efficiency of miRNA silencing from a mechanistic perspective - though I understand that there is a clear and solid correlation with MTS function. The best way to address this would be an experiment - knocking out dsRBPs or nuclear RBPs and checking for MTS access/efficiency. An alternative explanation would be that the sites that nuclear RBPs occupy are occupied in the cytoplasm by other RBPs (the authors surely noticed that from Bind'n'seq experiments some nuclear and cytoplasmic RBPs share related sequence motifs). Possibly also some dsRBPs share binding sites with ssRBPs in eCLIP - this would then mean that not *all* RBPs, but rather a subset function to help with miRNA targeting. The presence of overlapping binding sites between RBPs could and possibly should be considered.

Reviewer #3 (Remarks to the Author):

The study by Kim et al. has been substantially improved since the revision with extensive new analysis and experiments. However, I still have the concern that the authors may look at indirect upstream regulators of miRNA targeting. I am not convinced that all RBPs functions to open RNA structure to increase miRNA targeting. The following are my detailed comments on the revised texts:

1, the new analysis in Fig. 4e is nice but intriguing. Why a double-stranded binding protein, if not a helicase, functions to open RNA structure to facilitate miRNA targeting? In the list given by the authors, HNRNPU, TARDBP, YBX3 are more likely to be single-stranded binding protein, according to a recent study (Sun et al. Cell Research 2021), where the authors correlates RBP binding with RNA structure data and they found most so-called double-stranded binding proteins can also bind single-stranded RNAs. Would it be possible for the authors to repeat the analysis with the RBP annotations in that study to check the conclusion?

2, Even better, to repeat the analysis with the information that a certain binding site is single-stranded or double-stranded? More generally, to repeat the analysis by separating all RBP binding sites into more single-stranded or double-stranded according to RNA structure profiling data. This analysis lumped all RBP binding sites together and that is something bewildered me. I strongly think the analysis should be performed on the level of RNA structure and that is more directly relevant to miRNA targeting changes.

3, Again, the study used “preserved RBP sites” and assume them to be stable in other cell lines. This is better than the original submission but still not satisfactory. The study by Sun et al. Cell Research 2021 predicted RBP sites using measured RNA structural information in matched cellular context. Can the authors use the prediction from that study to repeat the analysis?

4, I once suggested to profile the RNA structures in wildtype of a knock-out of a certain RBP to directly see how the role of RNA structure in miRNA binding. I just found that the study by Spitale et al. Nature 2015 had such a dataset – they have already profiled RNA structure changes in Mettl3^{-/-} cells. Can the authors repeat the analysis by separating the lost Mettl3 binding sites according to RNA structure changes and see whether RNA structure change has a direct correlation with miRNA binding?

5, around lines 140, the author performed the analysis is DMS-seq data. Can they repeat the analysis with icSHAPE data, which has higher coverage and less bias from A and C nucleotides, as claimed in Spitale et al. Nature 2015.

6, lastly, I am always confused with the RNA structure change as a signal of miRNA binding change. On one hand, opening of RNA structure facilitate miRNA binding; but on the other hand, increase of miRNA binding will make an RNA looks like double-stranded. Which will win when you measure RNA structure at a miRNA-targeting-changing site?

Response to reviewers' comments

We would like to thank the reviewers for their careful and thorough evaluation of this manuscript and for the insightful comments and constructive suggestions, which helped us to improve the quality of the manuscript. Please see below, in blue, our detailed response to reviewers' comments.

Reviewer #1

(R1-1) In the revised manuscript, the authors have addressed some of my initial concerns and improved the clarity. However, it is still quite convoluted. It is important to note that the average effect for the strongest effect (overlapped or 1~20nts) was $\sim 2^{-0.18}$, roughly 12% downregulation, whereas for the weakest effect (1~10k) was $2^{-0.07}$, roughly 5% downregulation (**Fig. 1b**). Although these results may be statistically significant, it is unclear how many of individual cases have measurable effect. I caution that it is inappropriate to extrapolate the conclusion to all of >1,500 RBPs, based on these analyses. I also suggest the authors change the very strong claim throughout the manuscript, such as “widespread regulatory impact of >1,500 RBPs on MT” in the abstract; “the observed effect of RBPs on MT is undoubtedly a phenomenon occurring in an endogenous environment” **pg.5 line 105**.

As the reviewer pointed out, the difference in miRNA targeting (MT) efficacies might seem to be modest at first: it ranges from 8 to 11% in natural scale when comparing a group of genes with miRNA target site (MTS) overlapped with RBP-binding site (RBS) (**Fig. 1b, the leftmost bin**) and a group of genes with the longest distance between MTS and RBS ($d_{\text{MTS-RBS}}$) (**Fig. 1b, the rightmost bin**). Although miRNAs are widely accepted to mainly function as a fine tuner due to their modest repression of target mRNA level, miRNAs exert a significant impact on the regulation of thousands of target genes involved in various biological pathways in a collective and combinatorial manner. For example, in our previous study (Baek *et al.*, 2008, Nature), we have shown that the deletion of miR-223 exhibited 9% of subtle target mRNA derepression on average and yet this 9% derepression was sufficient to induce dramatic change in phenotype, a hundreds-fold overproduction of mouse neutrophils (Johannidis *et al.*, 2008, Nature). This example readily illustrates what we should focus on is the relative ratio of MT efficacies between comparing gene groups rather than the absolute value of the difference in repression. Therefore, in the miRNA field, 2- to 4-fold stronger repression of the genes with MTSs overlapped with RBS (**Fig. 1b, the leftmost bin**) compared to the genes with the longest $d_{\text{MTS-RBS}}$ (**Fig. 1b, the rightmost bin**) is regarded as a striking difference, which may well be sufficient to have physiological impact.

To assess the number of individual cases that have a measurable effect by miRNA targeting, we have defined multiple cutoffs for measurable effect. The number of individual cases and the percentage of their occurrences at the cutoffs of the $\log_2(\text{mRNA fold change}) < -0.3$ (19% downregulation), -0.5 (29% downregulation), and -1.0 (50% downregulation) was calculated. Our result indicates ~30% of overlapped cases (**the leftmost bin**) show >19% downregulation at the cutoff of -0.3 (**Fig. R1-1a**). When comparing with that of the longest $d_{\text{MTS-RBS}}$ group (**the rightmost bin**), we observed 2~3-fold increase of the individual cases at the cutoff of -1.0 (**Fig. R1-1b**).

We agree with the reviewer that the several claims are too strong and have revised these statements in our manuscript by removing the extrapolation to all of >1,500 RBPs and experimentally unvalidated claims. Thanks for the reviewer's thoughtful suggestion, we have revised our manuscript to improve the clarity of our manuscript.

Figure R1-1. The percent of individual cases that have a measurable effect (related to Fig. 1b).

a, The number of individual cases with measurable effect when applying various cutoffs for the repression.

b, Fold increase by comparing the percent of measurable cases in the leftmost bin (overlapped cases) with that in the rightmost bin (with the longest distance of $d_{\text{MTS-RBS}}$).

(R1-2) To further improve the clarity, the authors should clearly explain exactly which miRNAs and RBPs are used to generate many panels such as **Fig. 1b, 1d, 2d**, etc., and how the number of observations were calculated. They should also use individual miRNAs (2-5 of them) to generate the panels similar to **1b**, and examine whether the general observation still holds true for single miRNA.

For the analysis of **Figure 1b**, we used all RBPs downloaded from ENCODE project. The list of used miRNAs and RBPs is provided in **Supplementary Table 2** and **Figure 2a-c**. The number of observations was calculated by following the process described in **Supplementary Discussion**. Briefly, we defined each measurement of the $d_{\text{MTS-RBS}}$ as a single observation and the observations were counted for each combination of miRNAs and RBPs. In the analysis of **Figures 1d and 2d**, we selected a subset of the observations that have statistically indistinguishable confounding features among subgroups.

Following the reviewer's suggestion, we have repeated the analysis similar to **Fig. 1b** for individual miRNAs. As shown in the examples of **Figure R1-2**, association between $d_{\text{MTS-RBS}}$ and MT efficacy was consistently observed in most of each transcriptome data; overall, 162 out of 170 transcriptome data showed significant positive association (Pearson's correlation $P < 0.05$). This result indicates our observed association between RBP binding and enhanced MT holds true at the level of individual miRNAs as well.

Figure R1-2. The effect of individual miRNAs on MT efficacy.

Association analysis between $d_{\text{MTS-RBS}}$ and MT efficacy for individual miRNAs. From each of HepG2, HeLa, and other human cancer cell lines (OHCC) datasets, a randomly selected transcriptome data was displayed as an example. Sequence of siR-50 is CGCAAGUCUCCAACAUGCCUU (**Supplementary Table 2**).

Reviewer #2

The authors comprehensively engaged with all reviewers' comments and managed to convincingly alleviate most of the concerns.

(R2-1) Their basic observation that increased RBP binding around MTS is correlated with higher efficiency of miRNA silencing appears solid. The explanation that RBP binding prevents RNA secondary structure formation and thus enhanced access to the MTS site is convincing and the likely explanation. Nevertheless, I find that there is no convincing explanation offered as to why dsRBPs or nuclear RBP binding would result in increased efficiency of miRNA silencing from a mechanistic perspective - though I understand that there is a clear and solid correlation with MTS function. The best way to address this would be an experiment - knocking out dsRBPs or nuclear RBPs and checking for MTS access/efficiency. An alternative explanation would be that the sites that nuclear RBPs occupy are occupied in the cytoplasm by other RBPs (the authors surely noticed that from Bind'n'seq experiments some nuclear and cytoplasmic RBPs share related sequence motifs). Possibly also some dsRBPs share binding sites with ssRBPs in eCLIP - this would then mean that not *all* RBPs, but rather a subset function to help with miRNA targeting. The presence of overlapping binding sites between RBPs could and possibly should be considered.

The reviewer raised a concern that some binding sites of the dsRBPs or nuclear RBPs would also be occupied by ssRBPs or cytoplasmic RBPs, and this overlap could be the reason why we observed an association between the enhanced MT and the binding of dsRBPs or nuclear RBPs near MTs.

Although knocking out dsRBPs or nuclear RBPs would be helpful to elucidate the mechanism of how binding of dsRBPs or nuclear RBPs enhance MT, it is beyond the scope of our study, and the construction of another RBP KO cell line and performing the necessary experiments would take several additional months. Therefore, we addressed this issue with revised analyses which additionally consider the overlap between RBSs, as also suggested by the reviewer.

We calculated the fraction of dsRBP-binding sites overlapped with binding sites of other RBPs (**Fig. R2-1a**), and repeated the association analysis with subsets of the RBPs that are grouped based on their overlapping fraction (**Fig. R2-1c**). Consistent with the result of the dsRBPs with a high fraction of overlapping sites, binding of dsRBPs with a low fraction was clearly associated with enhanced MT (**Fig. R2-1c**). Also, for the nuclear RBPs, the association was consistently observed, regardless of the fraction of overlapping binding sites with cytoplasmic RBPs (**Fig. R2-1b, d**). These results indicate that our observed association between enhanced MT and the binding of the dsRBPs or nuclear RBPs near MTs are not biased by binding of other ssRBPs or cytoplasmic RBPs. As we described in the first revision, we are postulating the ssRNA-binding activity of dsRBPs and the presence of small amounts of nuclear RBPs in the cytoplasm seems to contribute to enhancing MT, and its detailed mechanisms are worth investigating in future studies. Thanks to the reviewer, we have improved our result by more precisely dissecting the impact of dsRBPs and nuclear RBPs binding on MT, and have added this result into our manuscript (**Supplementary Fig. 6**).

Figure R2-1. Association analysis between MT efficacy and binding of double-stranded RNA binding proteins (dsRBPs) or nuclear RBPs.

a, b, Relative fraction of the RBP-binding sites (RBSs) of dsRBPs (**a**) or nuclear RBPs (**b**) overlapped with RBSs of other RBPs. For more details, see **Supplementary Methods**.

c, d, Association analysis with dsRBPs (**c**) or nuclear RBPs (**d**) with low or high fraction of overlap with other RBPs. The mean mRNA fold changes of 3'UTRs obtained from mRNA-seq datasets that monitored the whole-transcriptome response to ectopically introduced miRNAs were shown. Each subgroup was carefully chosen to have statistically indistinguishable features, such as the four known confounding features and the RBP-binding intensity of the nearest RBS among the four subgroups.

Reviewer #3

Thank you for the consideration of our manuscript. While addressing the comments given by Reviewer #3 in the second revision, we have noticed potentially serious problems in the comments of Reviewer #3, which can be categorized as follows.

- I. **Newly added suggestion:** These comments present novel topics that are largely unrelated to our first revision.
- II. **Already addressed question in our previous response:** These comments are asking to perform analyses that are already addressed in our first revision.
- III. **Time-consuming and infeasible suggestion:** The suggested analyses will take several additional months to be completed while the results will not change the conclusion of our study.
- IV. **Irrelevant and beyond the scope of our study:** These comments are irrelevant and beyond the scope of our study. For instance, Reviewer #3 is suggesting to utilize a published dataset from a mouse cell line, that cannot be combined with the ENCODE dataset (human cell lines).

We believe that numerous comments given by Reviewer #3 in the second revision are inappropriate and unreasonable, which is far from constructive scientific review process. Therefore, we would like to ask that these comments by Reviewer #3 be excluded when making the editorial decision.

The study by Kim *et al.* has been substantially improved since the revision with extensive new analysis and experiments. However, I still have the concern that the authors may look at indirect upstream regulators of miRNA targeting. I am not convinced that all RBPs functions to open RNA structure to increase miRNA targeting. The following are my detailed comments on the revised texts:

(R3-1) The new analysis in **Fig. 4e** is nice but intriguing. Why a double-stranded binding protein, if not a helicase, functions to open RNA structure to facilitate miRNA targeting? In the list given by the authors, HNRNPU, TARDBP, YBX3 are more likely to be single-stranded binding protein, according to a recent study (Sun *et al.* Cell Research, 2021), where the authors correlates RBP binding with RNA structure data and they found most so-called double-stranded binding proteins can also bind single-stranded RNAs. Would it be possible for the authors to repeat the analysis with the RBP annotations in that study to check the conclusion?

Categories I and II – We already explained the potential mechanism of how the binding of dsRBPs enhances MT efficacy by its dual capability to bind both ssRNAs and dsRNAs (**R3-1 in our first revision and Supplementary Information**).

(R3-2) Even better, to repeat the analysis with the information that a certain binding site is single-stranded or double-stranded? More generally, to repeat the analysis by separating all RBP binding sites into more single-stranded or double-stranded according to RNA structure profiling data. This analysis lumped all RBP binding sites together and that is something bewildered me. I strongly think the analysis should be performed on the level of RNA structure and that is more directly relevant to miRNA targeting changes.

Categories I and II – We already performed DMS-seq in HEK293T WT and *IGF2BP1* KO cell lines (**R3-7**

in our first revision and Fig. 5e) and analyzed AGO2 PAR-CLIP-seq data (Fig. 3e) to directly show RBP binding opens the RNA secondary structure and enhances miRNA targeting.

(R3-3) Again, the study used “preserved RBP sites” and assume them to be stable in other cell lines. This is better than the original submission but still not satisfactory. The study by Sun *et al.* Cell Research, 2021 predicted RBP sites using measured RNA structural information in matched cellular context. Can the authors use the prediction from that study to repeat the analysis?

Categories I and III – This analysis would take several additional months to complete while the results will not change the conclusion of our study. Furthermore, the stability of our preserved RBP sites is validated by the consistency of our results in various cell lines; HepG2, HeLa, Other human cancer cell lines (OHCCs) (Figs. 1 and 2), and HEK293T (Fig. 6c).

(R3-4) I once suggested to profile the RNA structures in wildtype of a knock-out of a certain RBP to directly see how the role of RNA structure in miRNA binding. I just found that the study by Spitale *et al.* Nature, 2015 had such a dataset – they have already profiled RNA structure changes in *Mettl3*^{-/-} cells. Can the authors repeat the analysis by separating the lost *Mettl3* binding sites according to RNA structure changes and see whether RNA structure change has a direct correlation with miRNA binding?

Categories II and IV – We already performed DMS-seq in HEK293T WT and *IGF2BP1* KO cell lines (R3-7 in our first revision and Fig. 5e) and the recommended *Mettl3*^{-/-} data is only available for mouse.

(R3-5) Around lines 140, the author performed the analysis is DMS-seq data. Can they repeat the analysis with icSHAPE data, which has higher coverage and less bias from A and C nucleotides, as claimed in Spitale *et al.* Nature, 2015.

Categories III and IV – We already showed DMS-seq is sufficient to validate our claim that RBP binding opens the RNA secondary structure (Fig. 3b-d and 5e). Besides, the recommended icSHAPE data is only available for mouse and therefore cannot be combined with the ENCODE dataset (human cell lines).

(R3-6) Lastly, I am always confused with the RNA structure change as a signal of miRNA binding change. On one hand, opening of RNA structure facilitate miRNA binding; but on the other hand, increase of miRNA binding will make an RNA looks like double-stranded. Which will win when you measure RNA structure at a miRNA-targeting-changing site?

Categories I and IV – This is a newly added comment that is irrelevant to the first round of revision and beyond the scope of our study.

REVIEWERS' COMMENTS

Reviewer #1 (Remarks to the Author):

The authors have satisfactorily addressed my concerns. I have no more comments.

Reviewer #2 (Remarks to the Author):

In the revised manuscript, the authors engaged with my concern whether the observed effect of dsRBPs and nuclear RBPs indeed enhance miRNA targeting, or whether this may be an effect of overlapping target sites for those classes of RBPs with cytoplasmic RBPs. They refine their analysis by separating dsRBPs and nuclear RBPs based on the degree of target site overlap with other RBPs into those that seem to have little or lot of overlap. They then go on to show that there is little difference. However, this analysis is not really answering the question because a) eCLIP data are not saturated and thus the real overlap may be underestimated in some cases (the quality of some eCLIP datasets is quite limited and thus degrees of overlap may be over- or underestimated) and b) even the "low overlap" class still shows ~60% of binding sites overlapping, and it could be precisely those 60% of sites that may be contributing to the observed effect on miRNA targeting.

Fundamentally, the question remains as to why nuclear RBPs could enhance miRNA targeting in the cytoplasm. Research by the Bartel lab and others clearly indicates that miRNA targeting is a stoichiometric problem and to influence this process equivalent numbers of molecules are needed in the compartment that miRNA targeting takes place. If the authors persist in trying to make the argument that virtually all RBPs influence miRNA targeting they need to address the question in a plausible manner as to why RBPs that don't come into contact with the miRNA machinery physically still influence the process. The Supplementary Discussion of this problem was not convincing or sufficient.

Response to reviewers' comments

Reviewer #1

The authors have satisfactorily addressed my concerns. I have no more comments.

Thanks for your thoughtful review.

Reviewer #2

In the revised manuscript, the authors engaged with my concern whether the observed effect of dsRBPs and nuclear RBPs indeed enhance miRNA targeting, or whether this may be an effect of overlapping target sites for those classes of RBPs with cytoplasmic RBPs. They refine their analysis by separating dsRBPs and nuclear RBPs based on the degree of target site overlap with other RBPs into those that seem to have little or lot of overlap. They then go on to show that there is little difference. However, this analysis is not really answering the question because a) eCLIP data are not saturated and thus the real overlap may be underestimated in some cases (the quality of some eCLIP datasets is quite limited and thus degrees of overlap may be over- or underestimated) and b) even the "low overlap" class still shows ~60% of binding sites overlapping, and it could be precisely those 60% of sites that may be contributing to the observed effect on miRNA targeting.

Fundamentally, the question remains as to why nuclear RBPs could enhance miRNA targeting in the cytoplasm. Research by the Bartel lab and others clearly indicates that miRNA targeting is a stoichiometric problem and to influence this process equivalent numbers of molecules are needed in the compartment that miRNA targeting takes place. If the authors persist in trying to make the argument that virtually all RBPs influence miRNA targeting they need to address the question in a plausible manner as to why RBPs that don't come into contact with the miRNA machinery physically still influence the process. The Supplementary Discussion of this problem was not convincing or sufficient.

We agree with the reviewer to some degree that it would be an overstatement to claim almost all dsRBPs and nuclear RBPs also enhance miRNA targeting based on our current analyses. To alleviate reviewer's concerns, we have discussed the caveats of our current study and revised our **Discussion and Supplementary Discussion**. We stated that our data should be interpreted cautiously because the eCLIP-seq dataset is currently limited and therefore we were unable to completely separate dsRBPs from ssRBPs and nuclear RBPs from cytoplasmic RBPs. We would like to thank the reviewer for these insightful comments.